# A decade of detailed observations (2008–2018) in steep bedrock permafrost at Matterhorn Hörnligrat (Zermatt, CH)

Samuel Weber[1,2,3], Jan Beutel[2], Reto Da Forno[2], Alain Geiger[4], Stephan Gruber[5], Tonio Gsell[2], Andreas Hasler[6], Matthias Keller[2], Roman Lim[2], Philippe Limpach[4], Matthias Meyer[2], Igor Talzi[7], Lothar Thiele[2], Christian Tschudin[7], Andreas Vieli[1], Daniel Vonder Mühll[8], and Mustafa Yücel[2]

[1]Department of Geography, University of Zurich, Switzerland
[2]Computer Engineering and Networks Laboratory, ETH Zurich, Switzerland
[3]Chair of Landslide Research, Technical University of Munich, Germany
[3]Department of Geography, University of Zurich, Switzerland
[4]Institute of Geodesy and Photogrammetry, ETH Zurich, Switzerland
[5]Carleton University, Ottawa, Canada
[6]SensAlpin GmbH, Davos, Switzerland
[7]Computer Science Department, University of Basel, Switzerland
[8]SystemsX.ch Management Office, ETH Zurich, Switzerland

**Correspondence:** Samuel Weber (weber.samuel@tum.de)

**Abstract.** The PermaSense project is an ongoing interdisciplinary effort between geo-science and engineering disciplines and started in 2006 with the goals to realize observations that previously have not been possible. Specifically, the aims are to obtain measurements in unprecedented quantity and quality based on technological advances. This paper describes a unique ten+ year data record obtained from in-situ measurements in steep bedrock permafrost in an Alpine environment on the Matterhorn

5   Hörnligrat, Zermatt Switzerland at $3500\,\mathrm{m}$ a.s.l.. Through the utilization of state-of-the-art wireless sensor technology it was possible to obtain more data of higher quality, make this data available in near real-time and tightly monitor and control the running experiments. This data set (DOI: doi.pangaea.de/10.1594/PANGAEA.897640, Weber et al., 2019) constitutes the longest, densest and most diverse data record in the history of mountain permafrost research worldwide with 17 different sensor types used at 29 distinct sensor locations consisting of over 114.5 million data points captured over a period of ten+

10   years. By documenting and sharing this data in this form we contribute to making our past research reproducible and facilitate future research based on this data e.g. in the area of analysis methodology, comparative studies, assessment of change in the environment, natural hazard warning and the development of process models. Finally, the cross-validation of four different data types clearly indicates the dominance of thawing-related kinematics.

## 1   Introduction

15   The behavior of frozen rock masses in steep bedrock permafrost rock slopes is a dominant factor influencing slope stability when permafrost warms or thaws (Fischer et al., 2006; Ravanel and Deline, 2014). Ongoing degradation of mountain permafrost coincides with observations of increasing rockfall activity (Ravanel and Deline, 2011; Huggel et al., 2012; Gobiet et al., 2014) potentially triggering large scale hazard events via complex process chains (Huggel et al., 2005; Westoby et al., 2014; Haeberli

et al., 2017). While the long-term trend of rising permafrost temperatures can clearly be observed at a global scale (Biskaborn et al., 2019) warming trends of mountain permafrost are more diverse in their behavior (Noetzli et al., 2018). For example it has been recently observed that the generally warming trend can be temporarily interrupted depending on the amount and temporal extent of the snow cover (Noetzli et al., 2019) which is especially variable in mountainous terrain.

5     Numerous studies investigated the thermal and mechanical properties of frozen rock (e.g. Mellor, 1973; Davies et al., 2001; Gruber et al., 2003; Sass, 2004; Gruber et al., 2004b; Sass, 2005; Krautblatter and Hauck, 2007; Günzel, 2008; Gischig et al., 2011; Krautblatter et al., 2013; Jia et al., 2015; Mamot et al., 2018) with the goal of furthering our understanding of the processes acting in bedrock permafrost in the short- and long-term (e.g. Walder and Hallet, 1985; Wegmann, 1998; Hall et al., 2002; Murton et al., 2006; Matsuoka and Murton, 2008; Hasler et al., 2011a; Girard et al., 2013; Draebing et al., 2017). Several 10  studies highlighted the relevance of dense, diverse and long-term monitoring (Hasler et al., 2011b, 2012; Weber et al., 2017) in order to mitigate effects of temporal (annual) variability and other measurement artifacts (outliers) with negative impacts on data quality and therefore potentially leading to misinterpretation (Weber et al., 2018c, b).

    The Matterhorn Hörnligrat field site located in Zermatt, Switzerland at $3500\,\mathrm{m}$ a.s.l. is a unique situation for steep bedrock permafrost research as it is located on a ridge and not on a mountain top or in a large rock face where permafrost boreholes 15  would typically be placed (Luethi and Phillips, 2016). A comprehensive multi-sensor setup has enabled research on surface processes and kinematics in steep bedrock permafrost in the context of environmental forcing (ambient meteorological conditions, snow cover, heat flux) since 2006. Situated in a unique and iconic setting, the Matterhorn Hörnligrat field site now provides over a decade of mountain permafrost data: the longest, densest and most diverse data record with respect to permanent monitoring of mountain permafrost at high elevation worldwide. Apart from duration and location, this data set is novel 20  with respect to the diversity of the instruments used (17 different sensor types are contained in this paper), the density of the measurements both spatially (sensors are installed at 29 distinct sensor locations each containing one or more sensor types (see Table 2) and temporally (sampling rates on the order of per-minute to per-second). The data set presented amounts to $83.8\,\mathrm{GB}$ of data in 41'031 files of different formats containing approximately $114.4 \times 10^{6}$ data points of primary and aggregated data (see Table 6). To the best of our knowledge, in the entire European Alps only the Aiguille du Midi site (Chamonix, 25  France, $3842\,\mathrm{m\,a.s.l.}$) (Ravanel and Deline, 2011; Magnin et al., 2015), the permafrost borehole on Jungfraujoch (Grindelwald, Switzerland, $3700\,\mathrm{m\,a.s.l.}$) (Wegmann, 1997, 1998; Noetzli et al., 2019), the geothermal profiles on Stockhorn (Gruber et al., 2004c) and two simple ground surface temperature sensors located on the summit of Matterhorn ($4478\,\mathrm{m\,a.s.l.}$) are located in comparable or at higher altitude and are being operated in a long-term monitoring mode albeit the data records are shorter and offer less diversity with respect to the measurements. Other study sites at very high altitude exist, e.g. Grandes Jorasses 30  (Chamonix, France, $4208\,\mathrm{m\,a.s.l.}$) (Faillettaz et al., 2016) but have only been operated for a short period and in campaign mode. Outside of the European Alps, mountain permafrost data is very sparse (even in the Himalaya, Gruber et al., 2017) and in cases where ground-based measurements exist they are likely limited to a single sensor type only (Zhao et al., 2010; Popescu, 2018; Gruber et al., 2015).

    This manuscript documents the complete raw data at full sampling rates of the instruments used (primary data set, see 35  Section 3) for the most significant sensor channels/types deployed (as outlined in Section 4) as well as a selection of derived

data products (secondary data set). The derived data products are downsampled and cleaned time series of the weather station, ground temperature, electrical resistivity of rock, fracture displacement and inclinometer data as well as GNSS daily positions computed using double differencing techniques. In order to be able to fully understand and leverage the high-fidelity sensor and to allow full transparency and reproducibility a technology excerpt as well as the procedures for compiling and validating the primary and secondary data sets are presented in Section 3 and Section 5 respectively. Using the toolset described in Section 7 these data sets can be recreated and independently updated (living data process). The online data portal at http://data.permasense.ch (see Figure B1) is discussed in Appendix B. In addition, select examples of the data as well as an overview of the scientific results based on data from this field site are discussed in Section 6.

## 2 Matterhorn Hörnligrat field site

The Matterhorn is prominently known due to its archetypical form, the famous climbing route up the Hörnligrat ridge (northeast ridge of Matterhorn) and its dramatic first ascent on July 14, 1865. This first successful alpine conquests on the Matterhorn were actually undertaken by researchers: first ascent led by Edward Whymper, a writer and landscape illustrator on assignment from an English publishing house as well as subsequently the second ascent by John Tyndall, a prominent multidisciplinary scientist of his time both accompanied by local guides and other companions. Nowadays, several hundreds to few thousands of mountaineers climb the Matterhorn via the Hörnligrat every year.

In the exceptionally warm summer of 2003 increased rockfall activity was observed in the entire Alps (Gruber et al., 2004a; Ravanel et al., 2017). An increasing interest into the thermal behavior of permafrost in steep topographies in these years (Gruber et al., 2004b, c) lead to a first simplified modelling study based on the Matterhorn (Noetzli et al., 2007). It soon became clear that such work would require substantial evidence from long-term, in-situ measurements to calibrate and validate such models accordingly as no other comparable data set existed. Additionally, the prominent rockfall activity observed motivated further research questions with respect to slope/rock wall stability, natural hazards (mitigation) and the susceptibility of nearby human infrastructure and urban environments to such hazards (Gruber and Haeberli, 2007; Fort et al., 2009; Ravanel et al., 2017, 2010).

On July 15, 2003 a single rock volume of approximately $1500\,\text{m}^3$ released from the Hörnligrat at $3500\,\text{m}$ a.s.l. (white arrow in Figure 1a, CH1903+ 617950/92168) uncovering bare ice in the failure plan (see red arrows in Figure 1b and c) (Hasler et al., 2012). Although insignificant on the scale of a mountain the age and size of the Matterhorn as a whole, this particular incident showed significant susceptibility on the human scale to the processes governing such rockfall: as this rockfall event occurred in the middle of the summer climbing season and directly affected the popular climbing route to the summit, it led to the evacuation of 84 climbers by helicopter, the temporary closure of the climbing route and other mitigation measures (Haeberli et al., 2015). Compact ice was observed on the surface of the detachment scar right after the rockfall event suggesting clefts be filled with ice. With respect to the research aspects it is this hazard event, the expectation that further (catastrophic) dynamics would likely follow and the significant ice infill that led to the selection and instrumentation of the first experiments

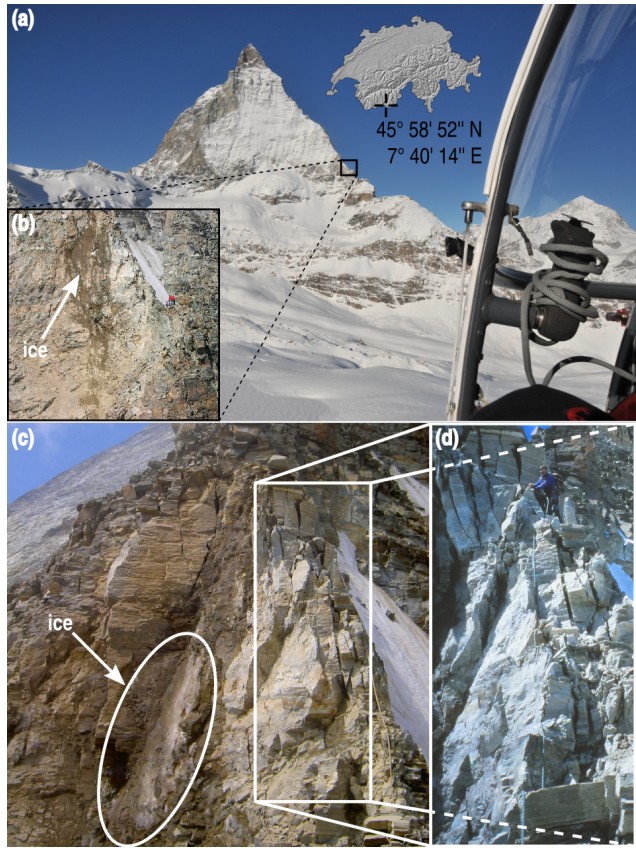

**Figure 1.** Matterhorn Hörnligrat field site is located on the North-East ridge of the Matterhorn on $3500\,\text{m}$ a.s.l. (b) and (c) show the detachment zone after the rockfall event in July 2003 with a volume of approximately $1000 - 2000\,\text{m}^3$. The comparison between (c) and the detail (d) taken 2-5 days before the rockfall event indicates that the top of the ridge was almost not affected by the failure event. Photos: PermaSense, Bruno Jelk and Kurt Lauber.

investigating kinematics of strongly fractured, steep bedrock permafrost in the years 2006-2007 at the Matterhorn Hörnligrat field site (Hasler et al., 2008).

Therefore an initial interdisciplinary project between geo-science and engineering was proposed with the initial goals to enable observations that previously have not been possible: the PermaSense project specifically aimed at (i) obtaining in-situ measurements with unprecedented quality and quantity (with respect to both spatial and temporal resolution and duration) but also (ii) to try to leverage then-emerging wireless sensor network technology (Talzi et al., 2007; Hasler et al., 2008) at scale and in a real case study. The Swiss Science Foundation (SNSF) funded National Competence Center on Mobile Information and Communication Systems (NCCR-MICS Aberer et al., 2007) as well as select government funding through the Swiss Federal Office of the Environment (FOEN) supported this initial push that over time development into a comprehensive outdoor

infrastructure and mountain lab supporting diverse experiments, long-term monitoring and online data: http://data.permasense.ch.

The Hörnligrat field site is located at $3500\,\text{m}$ on the North-East ridge of Matterhorn covering the area around the detachment zone of the 2003 rockfall event and consists of steep, highly fractured rock slopes with partially debris covered ledges and different expositions, where the expected occurrence of permafrost varies with aspect and relief conditions (see Figure 3 in Weber et al., 2017). Geologically, this field site consists of gneiss and amphibolite of the Dent Blanche nappe (Bucher et al., 2004) and the most dominant fractures are oriented parallel to the ridge and dip nearly vertical (see Figure 2 in Hasler et al., 2012). Climatically, the region of Zermatt is characterized by a dry and subcontinental climate with high daily/seasonal temperature fluctuations and with mean annual air temperature (MAAT) of $3.5\,°\text{C}$ (1961–1990) and $4.2\,°\text{C}$ (1981–1990) (MeteoSwiss, 2019). While reanalysis data with a $1\text{x}1\,\text{km}^2$ grid indicate a regional MAAT of $-6.7\,°\text{C}$ (1961-1990, Hiebl et al., 2009) for the field site area, local measurements at the field site show a mean annual air temperature of $-3.7\,°\text{C}$ (period 2011–2012, Weber et al., 2017). As precipitation mostly falls as snow with occasional infrequent rainfall events in summer, liquid water is mainly supplied to the site by snow melt (Hasler et al., 2012). Winter temperatures down to $-27\,°\text{C}$ in combination with exposure to strong wind (to over $100\,\text{km/h}$) results in a preferential snow deposition in fractures, on ledges and at other concave microtopographical features. While the northern side contains small ice field within a steep heterogeneous rock face, on the south side snow patches develop during winter in couloirs as well as on rock bands and disappear in spring/summer completely (Hasler et al., 2012).

Surveying and site selection took place in the years 2006/2007 with an initial sensor installation campaign in fall 2007 (Hasler et al., 2008). The technological developments started with data logger prototypes (Talzi et al., 2007) that were used for a first data retrieval campaign during the following winter season. The prototype development and initial experience resulted in a redesign of the wireless sensing platform that was deployed for the first time on July 25, 2008 (Beutel et al., 2009). This date also marked the start of the "production" data generation for the PermaSense project and the data contained in this publication. Later technological milestones include the introduction of the GSN data management system, a switch from 3G cellular connectivity to IEEE 802.11a $5\,\text{GHz}$ WLAN for long-haul connectivity and the introduction of a middleware software infrastructure for mitigating data loss through back-pressure in summer 2009. On the sensor side extensions took place in 2009 with a remote controlled high-resolution visible light camera (Keller et al., 2009b, a) as well as a significant extension of the crackmeters in summer 2010 and the installation of a high-precision survey-grade GNSS receiver at the very end of 2010 (Beutel et al., 2011). A local weather station was added in 2010 and a net total radiometer in 2015. After this first research phase focusing on prototyping and the investigation of surface kinematics with respect to thermal forcing (Hasler et al., 2012; Weber et al., 2017) an additional research avenue was added from 2012 onwards: a first pilot study using acoustic emission and based on similar efforts undertaken at the Jungfraujoch (Amitrano et al., 2012; Weber et al., 2012; Girard et al., 2012, 2013) aimed at characterizing damage evolution inside the solid rock walls in 2012/2013. A larger profiling experiment (Weber et al., 2018c) has been set up to investigate signals emanating from the mountain and possible damage events with different instruments ranging from $1\,\text{Hz}$ to $100\,\text{kHz}$ as well as additional L1-GPS measurement points starting in 2015/2016. Finally, in an effort to establish a vertical transect of thermal measurements spanning the whole mountain (two ground surface temperature

measurement points exist on the summit since 2011, maintained by Agenzia Regionale Protezione Ambiente Valle d'Aosta (ARPA VDA), Italy, permafrost boreholes maintained by PERMOS, SLF/WSL, Switzerland are located on lower elevations at the Hörnlihütte and Hirli) an extension with further ground surface temperature profiles implemented at $4003\,$m a.s.l. in the vicinity of the Solvay Hut higher up on the ridge has been performed. Despite its remoteness and exposure this field site is actually readily accessible being situated directly on and in the bottom segment of the climbing route with further infrastructure nearby (mountain hut, heliport, transportation facilities, Internet connectivity) and therefore can be accessed even in a day trip from Zurich.

After completing the first ten years since the first experiment went live in July 2008 it's now time to publish a first digest of this data including a thorough documentation in order to (i) preserve this data and (ii) make it available for future research in the broader context. The data presented in this publication constitutes a best-of of the most relevant and descriptive geo-science related data collected. There are further data available in the context of this work, that either (i) have been published elsewhere (Weber et al., 2018a; Meyer et al., 2018), (ii) is not deemed suitable for publication in the context of this publication (either out of scope or to complex or too poor in quality) and (iii) have been collected by related activities in the vicinity of this field site. The most relevant of these additional data sources are described in brief in Section 4.8 in order to give the reader the relevant pointers in this context.

## 3   Instrumentation technology and data management

The core instrumentation technology employed at this field site are autonomous, low-power wireless networked sensors (Beutel et al., 2009), frequently also referred to as wireless sensor network or short *sensor network*. The promise of this novel technology at the time of the conception of this field site in 2006-2007 (Hasler et al., 2008) was to allow unobtrusive, large-scale and highly reliable measurements based on a minimum resource footprint without a central point of failure and extensive cabling. Apart from geoscience investigations the first PermaSense project pursued the goal to develop means for long-term, high-quality sensing in harsh environments, generating better quality data, with online data access in near real-time (Hasler et al., 2008). Using such technology it would be possible to achieve measurements that previously have not been possible and consequentially to enable new science, answering fundamental questions related to decision making, natural hazard early-warning. For selected sensors, where the integration as low-power wireless sensor was infeasible or impractical, industry standard components have been used although they have typically been adapted and integrated with our custom network, data and power management infrastructure based on our sensor network technology. Our experience over the past decade+ shows, that using a WSN is a promising approach with superb data availability and data integrity. The sensor nodes have been running reliable and autonomous on the order of years in an extremely challenging environment and off-season/unplanned maintenance efforts are seldom necessary.

## 3.1 PermaSense low-power wireless sensing system

The PermaSense wireless sensor networks consists of Shockfish TinyNodes sensor nodes running the Dozer protocol stack (Burri et al., 2007) implemented in TinyOS (Levis et al., 2005). The sensor nodes are integrated on a custom Sensor Interface Board (Beutel et al., 2009) with power management, data acquisition, storage and interface protection functionality. The analog data acquisition frontend is built using a 16-bit resolution and 8-channel $\Sigma$-$\Delta$ analog to digital converter (Analog Devices AD7708) and an external precision voltage reference. The ADC is controlled by software running on the MSP430 micro-controller of the TinyNode. The data acquisition operation for both single-ended and differential measurements is configured with a static, periodic sampling rate strictly interleaving with networking operations, in our case $120\,\text{s}$. Other digital sensors, e.g. on-board system health, weather station, digital pressure and temperature sensors can be attached as well using a digital bus interface. The data from the sensor nodes is transferred using the Dozer ultra low-power multihop networking protocol stack (Burri et al., 2007). Data is forwarded to a central data sink, a base station, connected to the Internet with a period of $30\,\text{s}$. In cases of network congestion or loss of connectivity, e.g. due to excessive snow build up or base station failures, data is kept back on local storage on every node using a mechanism called backpressure. For this a $1\,\text{GB}$ non-volatile Flash memory storage (SD-card) is integrated on every node. With a power envelope of about $150\,\mu\text{A}$ these wireless sensors have been in continuous operation in the field for periods up to seven years based on a single D-size $\text{LiSOCl}_2$ cell (SAFT LSH-20, $13\,\text{A}\,\text{h}$), although due to maintenance and upgrading activities, in practice the typical operational time on location for a single node is shorter.

Similar to the backpressure mechanism on every sensor node, the base station also contains a local database for intermittent data storage in case connectivity to the database is lost. For reasons of power efficiency the sensor network does not support synchronization to absolute reference time (e.g. UTC) but relies on local 1-second granularity time keeping. The local times-tamp of every data sample generated on a sensor node is propagated through the Dozer network and based on the arrival time of each packet at the base station (a Linux system supporting time synchronization to a global reference) the *generation time* of the respective data sample is calculated using the method of "elapsed time of arrival" (Keller et al., 2012a). Since the forwarding network uses a dynamically changing topology it can happen that data is received out of order with respect to timing at the base station. Because of inevitable drifting behavior of all local clock sources and due to intermittent losses of end-to-end connectivity between nodes of the sensor network as well as on the TCP/IP networking segment slight jumps in the timing can occur (a detailed analysis of the network performance is available in Keller et al., 2011, 2012b). Nevertheless, these effects are not of concern with respect to the long-term nature of the processes observed (diurnal to seasonal behavior). For the user of this data it only matters that on accessing the online data streams on the online data portal in real-time, different timing information exists for every data sample referring to the estimated *generation time*, the time of arrival at the base station and the time of storage in the data base respectively and that very recent data may still be incomplete (out-of-order arrival with respect to time). Once data has been downloaded, quality checked and possibly also downsampled using the tools discussed in Section 3.4 and supplied alongside with the data in this paper, possible timing artifacts are no longer of concern.

## 3.2 Low data rate sensor integration

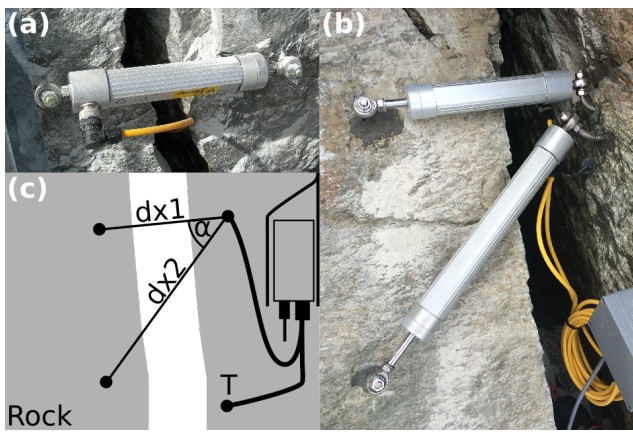

**Figure 2.** a) 1-axis and b) 2-axis crackmeter setup in the field. c) 2-axis crackmeter setup with one thermistor connected to the wireless sensor network. In cases where multiple crackmeters are mounted on a single location the angle $\alpha$ between the two crackmeters is given in combination with the length of the instrument.

The basic sensor used in combination with these wireless sensor nodes are temperature sensors (NTC thermistors) and fracture dilatation sensors (crackmeters) in different configurations ranging from single channel configurations to multiple channel configurations, e.g. 2x crackmeters and 1x thermistor (see Figure 2b) attached to a single wireless sensor node using 3x
single ended ADC channels, a half-bridge resistive divider with precision reference resistor and conversion after the Steinhart-Hart equation. A special configuration used are the rigid PermaSense sensor rod and thermistor chain (see Figure 3). These macro-sensor assemblies incorporate multiple thermistors as well as reference resistors, an internal multiplexer circuit allowing to sense at multiple locations (depths) simultaneously housed either in a rigid glass fiber reinforced tube (sensor rod) or located inside heat-shrink tubing and cable segments configured to length as desired. Two variants exist: (i) the original 12 mm 4-
channel sensor rod that additionally incorporates four electrode pairs allowing to measure resistivity at different depths and (ii) the revised 20 mm sensor rod that is designed without resistivity electrodes but rather in a configurable setup and using metal rings for better thermal coupling to the rock. Both configurations require a 1 m deep hole to be drilled. This most recent design is configurable with respect to the amount of sensors and the sensor depths allowing to manufacture assemblies that are compatible to commercially available units such as the UMS TH3 sensor rods that needed to be replaced as this unit is limited
in its measurement range below $-20\,^{\circ}\mathrm{C}$ and furthermore requires a lot of power to operate making it unsuitable for long-term monitoring.

Wireless L1-GPS sensor nodes equipped with an additional 2-axis inclinometer for the detection of terrain movement (Wirz et al., 2013) have been developed using the same principle as outlined above (Buchli et al., 2012). Only here GPS data, specifically the RAW output of the satellite observations constitutes the actual sensor data. Environmental forcing, e.g. ambi-

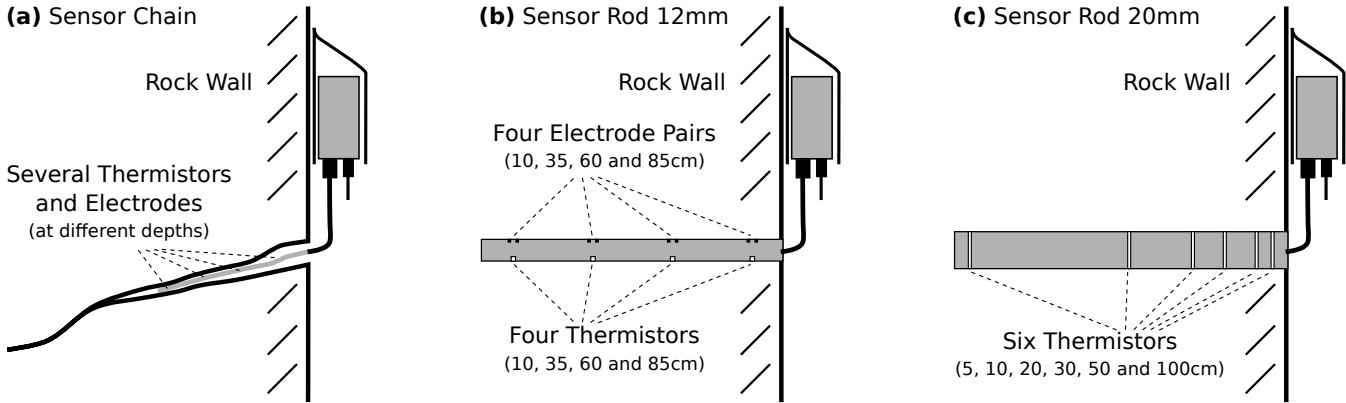

**Figure 3.** Sensor setup to measure temperature and resistivity in fractures and in rock.

ent weather conditions such as air temperature, wind or radiation are measured using commercial sensors (Vaisala WXT520 weather station and Kipp & Zonen CNR4 radiometer) integrated into the sensor network.

### 3.3 High data rate sensor integration

A number of sensors that are not suitable for integration in a low-power and low-data rate sensor network and that typically
come ready to deploy with a standard communication interface (e.g. USB, Ethernet) have been integrated into the field site as well. In order to minimize cabling these sensing systems (e.g. a DSLR camera, high-precision GNSS reference receiver, seismic data acquisition) have been integrated with a Wireless LAN router and facilities to monitor and control power (switch on/off both the sensing system and WLAN from remote). Using a mix of local and remote directional link-based WLAN connectivity between the Internet and the instruments on the field site is established based on a WLAN access point located at
the cable car station of the Klein Matterhorn 3883 m a.s.l. about $6.5\,\mathrm{km}$ away where the network is attached to a local Internet service provider using fiber.

### 3.4 Data management infrastructure

Care has been taken that all data collected are structured and stored in a coordinated fashion allowing reproducible research results and re-use of data in different contexts and in future projects. Also flexibility with respect to extensions (new sensor
types), support of different data rates, metadata integration and life-cycle management were taken into account. The data backend is implemented using a data streaming middleware where a dedicated processing structure called a virtual sensor is responsible for processing a specific data type, e.g. one virtual sensor for temperature measurements and another virtual sensor for images. Complete processing chains, can be implemented by concatenating virtual sensors either within the same instance of the Global Sensor Network (GSN Aberer et al., 2006) or also across multiple instances of GSN. In our case, data is processed
and stored in two concatenated instances of GSN: a private instance only accessible internally for primary, unprocessed data (green database instance in Figure 4) and a public instance for secondary, processed data and publishing this data via web

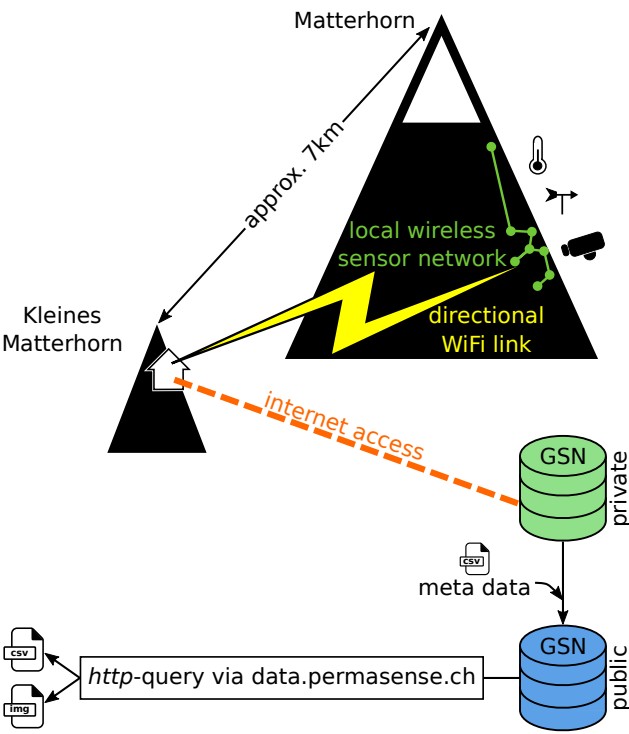

**Figure 4.** System architecture: data are collected with a local wireless sensor network and transmitted to the summit station of Kleines Matterhorn. The private GSN server receives the data, which are stored in a primary database. Data are passed on to a public GSN server where they are mapped to metadata (positions, sensor types, calibration, etc.) and converted to convenient data formats. Finally, data are available for download and analysis using external tools.

frontend to the user domain, i.e. the Internet (blue database instance in Figure 4). A visualization tool provides up-to-date key graphics (Keller et al., 2012a) on a web frontend where all all data can be accessed online at http://data.permasense.ch. Online data can be accessed using an Internet browser (see Figure B1) or using web queries (see Appendix B).

In this system all data of one specific data type and processing stage is kept in a single data structure with the virtual
5 sensor acting as its interface, i.e. all data of a specific type is kept in this respective data structure irrespective of time and location. The processing chains contain steps for the mapping of device IDs, sensor type and sensor IDs to positions for the respective time periods, applying the correct unit conversion functions according to the sensor type defined, decomposition of more complex data types (multiplexed data) into user-friendly data types and aggregation of data. Each instance of a virtual sensor is mapped to a unique data structure, e.g. a dedicated table on a MySQL database server. Data types with very large
10 amounts of (binary) data, e.g. images are stored directly on a networked file system and only a reference to the respective file is stored in the database. With this two-step data management pipeline consisting of a raw data ingress, dump and store in the first instance as well as multiple processing steps as outlined in the second instance it is assured that all data transactions are consistent, transparent, traceable and verifiable. Should corrections to the data be necessary, e.g. by inclusion of further

metadata, correction of metadata or the integration of alternate processing methods they can be applied by simply re-running the respective data from the private primary repository to the second instance with the modifications in place.

In order to consistently manage data of the field site a set of rules has been defined:

- An individual protocol sheet is used for each intervention (field work day) where all noteworthy items are recorded (installation, maintenance, removal)

- Sensor interventions on site take place at different times for each position. To simplify things, the whole day of an intervention is typically assumed as "invalid data".

- All sensor devices are mapped to a distinct position ID. The mapping contains to-from information, the device id (possibly MAC address), sensor type and calibration data.

- All data from a specific data source (sensor type) is kept in an individual data structure. Queries are typically made per data type and position ID.

- Detailed circumstances (crackmeter angles, thermistor depth) are recorded using auxiliary data formats: text files, excel files or photographs.

As described earlier the data ingress from the base station on the field site is based on a local database on the base station that allows to delay data transmission in cases of loss of connectivity or server outages. In the first years of the deployment this functionality did not yet exist and therefore a (then significant) data gap from June to August 2009 is visible in some of the thermal and crackmeter data due to a failure in the cooling system of the server room and a longer outage of the server system. With hindsight it must be said that this outage event, that had nothing to do with the actual field site instrumentation, exemplified in an extraordinary way the need for tight integration and synchronization of storage resources at all levels of a networked sensing system.

## 4 Detailed field site setup and description of the primary data products

This section gives an overview as well as details of the main sensor setup installed at Matterhorn Hörnligrat and describes the data provided with this paper. Table 1 provides an overview listing of the main sensors used grouped by sensor type including their approximate period of operation, units derived, measurement interval and key sensor characteristics. Table 2 and Figure 5 give a detailed listing of the location specific instrumentation detailing the number of sensing channels and sensor types available at each position. For every sensor type used, a detailed discussion of the specifics of each sensor type as well as installation and location specific information is given in the remainder of this section. Finally, Figure 6 gives a graphical overview of the data availability for all data products contained in this paper.

As described in Section 2 and also visible in Figure 6, the sensor setup at this field site has continuously grown over the years. There are only few data gaps. The data yield and reliability of the measurement systems has surpassed expectations. In a few

**Table 1.** Overview list of the sensors used ordered by sensor type.

| Sensor Type | Sensor | Period | Unit | Interval | Accuracy |
|---|---|---|---|---|---|
| Air temperature | Vaisala WXT520 | 12/2010 - ongoing | °C | 120 s | ±0.3 °C |
| Barometric pressure | Vaisala WXT520 | 12/2010 - ongoing | hPa | 120 s | ±1 hPa |
| Relative humidity | Vaisala WXT520 | 12/2010 - ongoing | %RH | 120 s | ±3 − 5 % RH |
| Wind speed | Vaisala WXT520 | 12/2010 - ongoing | km/h | 120 s | ±3 % at 10 m/s |
| Wind direction | Vaisala WXT520 | 12/2010 - ongoing | ° | 120 s | ±3° at 10 m/s |
| Precipitation | Vaisala WXT520 | 12/2010 - ongoing | mm | 120 s | resolution 0.01 mm |
| Radiation | Kipp & Zonen CNR4 | 06/2015 - ongoing | W/m$^2$ | 120 s | non-linearity <1 % |
| Ground temperature | PermaSense sensor rod 12 mm | 07/2008 - ongoing | °C | 120 s | ±0.2 °C |
| Ground temperature | UMS TH3 sensor rod 20 mm | 06/2015 - ongoing | °C | 120 s | ±0.1 °C |
| Ground temperature | PermaSense sensor rod 20 mm | 09/2017 - ongoing | °C | 120 s | ±0.1 °C |
| Ground temperature | Thermistors, YSI 44006 | 07/2008 - ongoing | °C | 120 s | ±0.2 °C |
| Ground resistivity | Custom copper electrodes | 07/2008 - ongoing | MΩ | 120 s | n.a. |
| Fracture displacement | Crackmeter Stump ForaPot | 07/2008 - ongoing | mm | 120 s | ±0.075 %, 5 ppm/°C |
| Time lapse photography | Nikon D300, 24mm f/2.8D | 08/2009 - ongoing | n.a. | 2 h | n.a. |
| L1/L2-GNSS observables; position coordinates | Leica GRX1200+ GNSS receiver, AR10 antenna | 12/2010 - ongoing | m | 30 s | n.a. |
| L1-GPS observables; position coordinates | L1 DGPS, u-blox LEA-6T, Trimble Bullet III antenna | 08/2014 - ongoing | m | 5 s, 30 s | n.a. |
| Inclination | Murata SCA830-D07 Inclinometer | 08/2014 - ongoing | ° | 120 s | ±30 mg |

cases (Position 2 – rockfall, Position 12 – sensor malfunctioning from initial installation) sensing positions have been retired but in general agreement exists that the sensor locations are well planned and selected and that the measurements obtained are representative for each respective location. For the sake of completeness it must be said that a few other sensor placements exist(ed) but due to their experimental nature and/or instability they are not part of this publication.

## 4.1 Weather station data

Since 2010 a local weather station based on a Vaisala WXT520 compact all-in-one weather instrument is installed on-site to obtain a more detailed weather data record comprising ambient air temperature (see black line in Figure 7), air pressure, relative humidity, wind (speed and direction) and precipitation. This has been extended with a 4-component net radiometer Kipp & Zonen CNR4 in the summer of 2015 (see green line in Figure 7 for shortwave radiation in). The net radiometer is installed without capabilities for ventilation and heating. The WXT520 is capable of heating the rain and wind sensor but for practical reasons this feature is only enabled when enough power is available which typically corresponds to good weather periods and turned off especially in prolonged bad-weather periods. Both instruments have been vendor calibrated and the

**Table 2.** Per position overview of sensor channels: for solid bedrock and fracture environments, the number of sensor channels are listed in this table.

| Position | Rock | | | | | Fracture | | | |
|---|---|---|---|---|---|---|---|---|---|
| | Temperature near-surface | Temperature 5 − 100 cm | Resistivity 5 − 100 cm | GNSS observables, coordinates | Inclination | Temperature near-surface | Temperature 5 − 300 cm | Resistivity 10 − 300 cm | Displacement |
| MH01 | 2 | | | | | | | | 1 |
| MH02 | 1 | | | | | | 3 | | 1|2[b] |
| MH03 | 2 | | | | | | 7 | | 1 |
| MH04 | | | | | | | 5 | | 1 |
| MH05 | | | | | | | 4 | 4 | |
| MH06 | 2 | | | | | | | | 2 |
| MH07 | | | | | | | 4 | 4 | |
| MH08 | | | | | | | | | 2 |
| MH09 | | | | | | 1 | | | 3 |
| MH10 | 1 | 4 | 10 | | | | | | |
| MH11 | 1 | 4|6[a] | 10|0[a] | | | | | | |
| MH12 | 1 | 4 | 10 | | | | | | |
| MH15 | | | | in situ radiometer | | | | | |
| MH18 | | | | | | | | | 1 |
| MH19 | | | | time lapse photography | | | | | |
| MH20 | | | | | | | | | 2 |
| MH21 | | | | | | | | | 2 |
| MH22 | | | | | | | | | 2|1[c] |
| MH25 | | | | in situ weather station[d] | | | | | |
| MH27 | | 6|6[a] | | | | | | | |
| MH30 | | 6 | | | | | | | |
| MH33 | | | | 1 | 1 | | | | |
| MH34 | | | | 1 | 1 | | | | |
| MH35 | | | | 1 | 1 | | | | |
| MH40 | | | | 1 | | | | | |
| MH42 / HOGR | | | | 1 | | | | | |
| MH43 | | | | 1 | | | | | |
| MH46 | | 6 | | | | | | | |
| MH47 | | 6 | | | | | | | |

[a] Intervention: change of sensor type. [b] Intervention: replacement and extension from 1-axis to 2-axes setup. [c] Intervention: one crackmeter broke due to rock fall. [d] Continuous sampling mode: 2016-12-01 – 2017-07-27 (but not 2017-06-28 – 2017-07-01), device change on 2018-09-15

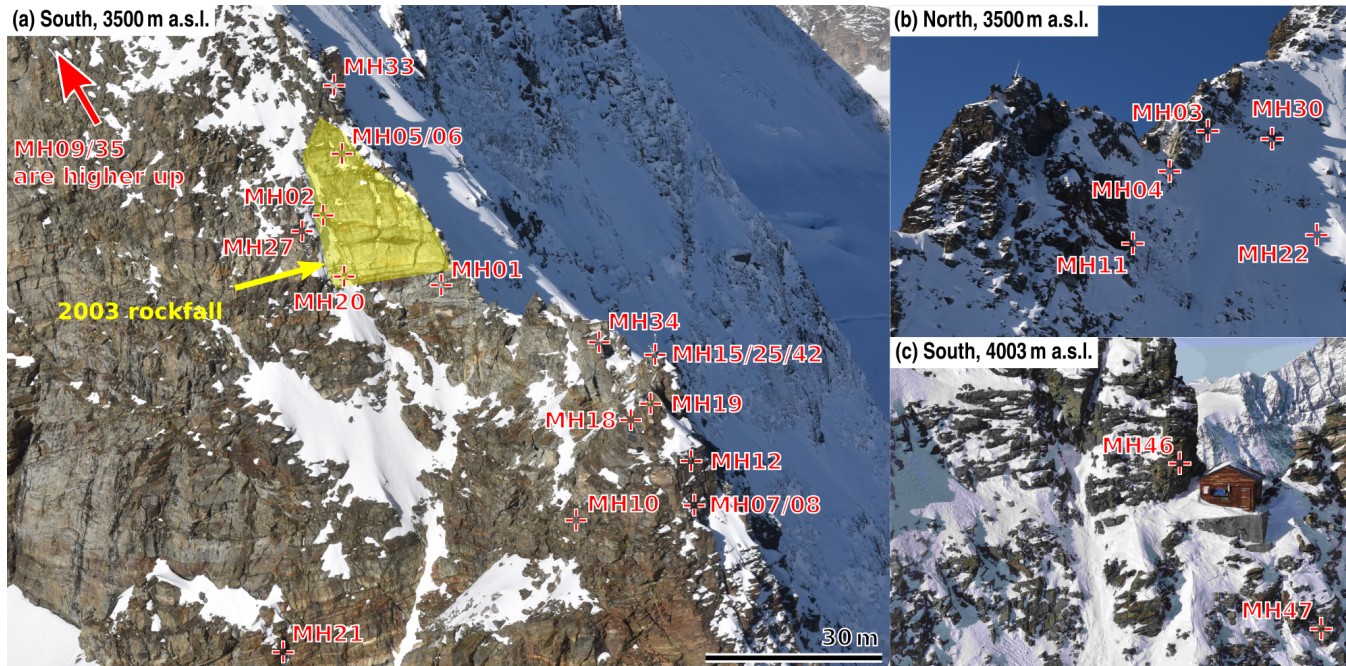

**Figure 5.** Overview of the field instrumentation at the Matterhorn Hörnligrat: a)+b) South and North side on approximately $3500\,\mathrm{m}$ a.s.l. next to the 2003 rock fall event. c) Extension next to the Solvay Hut on $4003\,\mathrm{m}$ a.s.l. with South exposition.

respective calibration data is applied in the data conversion procedures as advised by the manufacturer. It is well known that it is not straightforward to measure present weather conditions in such a hostile and exposed location, high up on the ridge of a $4000\,\mathrm{m}$ peak. Therefore this data must be treated with some caution. There are more data outages as with our other sensors. Clearly an instrument such as the Vaisala WXT520 designed to measure liquid precipitation (with the principle of counting and

integrating over the impacts of droplets on the sensor surface) is neither designed nor capable of measuring solid precipitation in any form. Further, the Vaisala WXT520 has been operated in different modes (interval vs continuous sampling) which resulted in different maximum/minimum wind velocity data. Also the application of a net radiometer on a high-alpine rock ridge is far from any WMO compliant sensor setup. Although in parts only indicative, the data obtained from these sensors is very valuable as it is local to the site and exhibits all the small scale local and temporal variability that regional models

extrapolating from national service weather data cannot capture, e.g. regular local cloud build up on the mountain slopes in the summer's late afternoons, detailed onset timing of local weather changes etc.

### 4.2   Ground temperature

Ground temperature data are recorded at different depths (ranging from near-surface, which refers to a depth of $3-8\,\mathrm{cm}$, to $3\,\mathrm{m}$ depth) inside fractures as well as in intact/solid rock. All measurement devices use NTC (Negative Temperature Coefficient)

thermistors potted in epoxy and calibrated with zero point calibration at $0\,^{\circ}\mathrm{C}$. Beside the single thermistor setup to measure

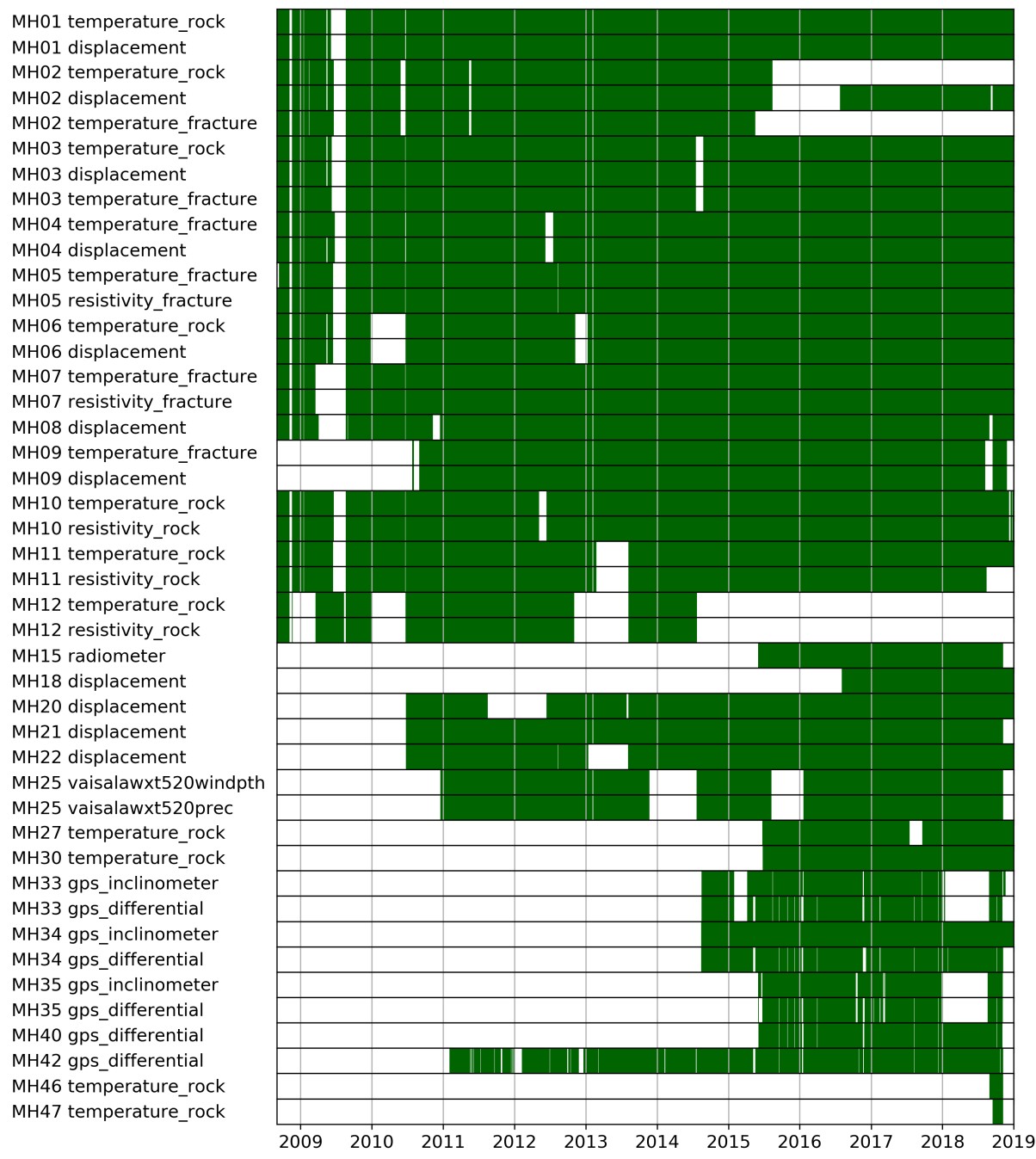

**Figure 6.** Data availability for all data products. The time periods when data is available are indicated in green.

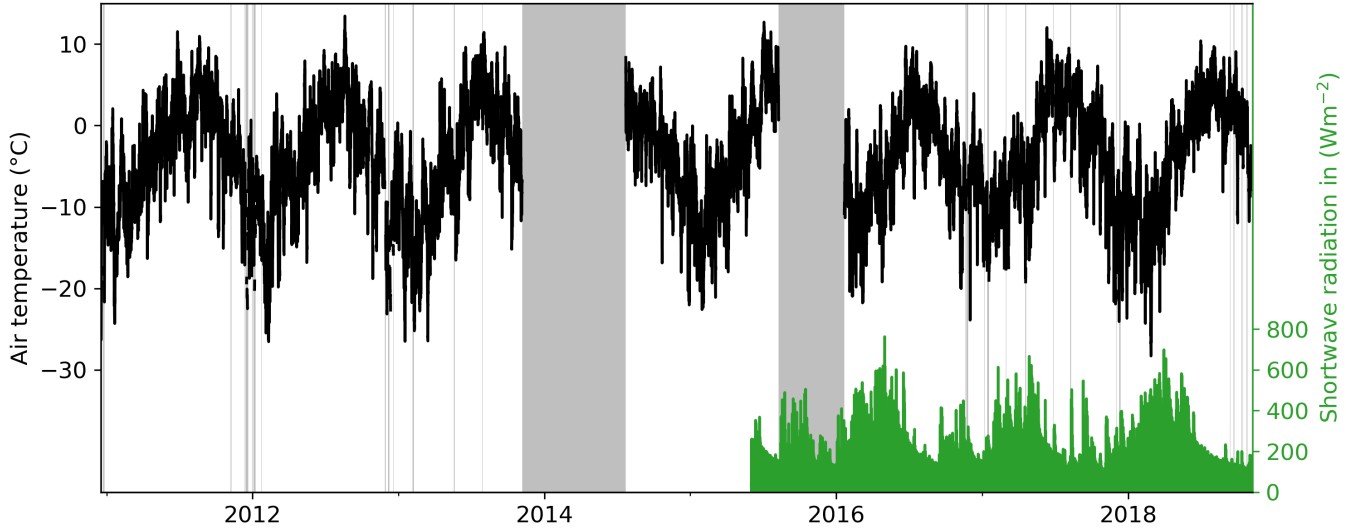

**Figure 7.** Air temperature and shortwave radiation data. Gray bars indicate data gaps.

near-surface temperature, two different major types are used to measure temperature at different depth: on the one hand sensor rods are drilled in the rock and on the other hand thermistor chains are deployed in fractures (see Table 1). All thermistor systems used have been calibrated using a single-point calibration scheme at $0\,°C$. The main characteristics of the four different temperature measurement devices used are given in the following:

1. PermaSense sensor rod $12\,mm$:

YSI-44006 NTC thermistors, interchangeable tolerance $\pm0.2\,°C$, Drift @ $0\,°C$ over 100 months <$0.01\,°C$

2. UMS TH3 sensor rod $20\,mm$:

Digital system with built in analog to digital converter (ADC). $\pm0.1\,°C$, measuring range $-20\,°C$ to $50\,°C$, resolution. $0.034\,°C$

3. PermaSense sensor rod $20\,mm$:

Measurement Specialities epoxy encapsulated 44031RC NTC thermistor mounted inside aluminum contact rings with thermally conductive epoxy, interchangeable tolerance $\pm0.1\,°C$ , Drift @ $0\,°C$ over 100 months <$0.01\,°C$

4. Various thermistor configurations (single or embedded in sensor chain):

YSI-44006 NTC thermistors, interchangeable tolerance $\pm0.2\,°C$, Drift @ $0\,°C$ over 100 months <$0.01\,°C$

Table 2 shows which temperature sensors are installed at which position, whereas Table 3 the depths of the thermistors. Figure 8 shows exemplary hourly rock temperatures measured at $10\,cm$ and $85\,cm$ depth and mean annual rock temperature at $85\,cm$ (MAGT_85cm) for years with more than $98\%$ data availability.

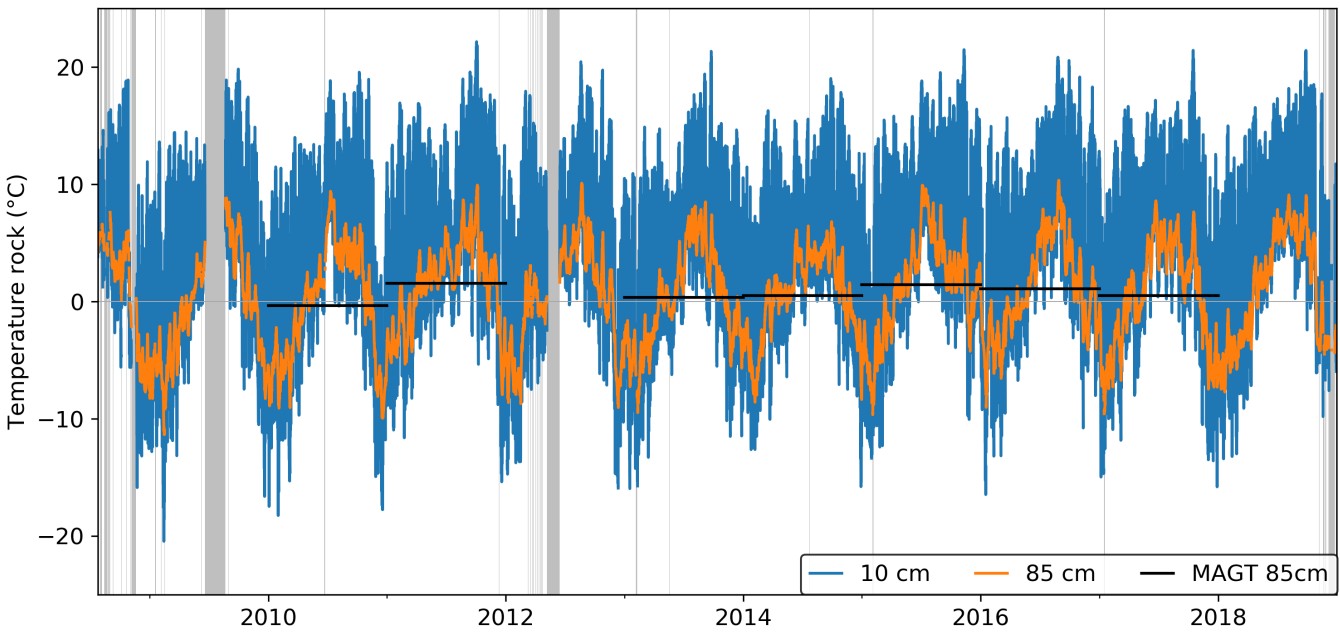

**Figure 8.** Rock temperature measured at different depths at position MH10. Black lines indicate mean annual rock temperature at $85\,\mathrm{cm}$ depth, if at least $98\%$ of the data are available. Gray bars indicate data gaps.

### 4.3 Ground resistivity

Electrical resistivity tomography (ERT) is a common geophysical method to characterize the shallow subsurface (Daily et al., 2012). ERT has successfully been used to observe temporal and spatial variations of moisture movement during freeze-thaw cycles in solid rock faces (Sass, 2004, 2005) and in solid permafrost rock walls in short- (Krautblatter and Hauck, 2007) and
5 long-term (Keuschnig et al., 2017) measurement campaigns.

The PermaSense sensor rod $12\,\mathrm{mm}$ is designed with four electrode pairs with a distance of a centimeter each that couple with the rock electrically using conductive foam pads (see Figure 3). In contrast to ERT-surveys, here the contact resistance is directly added to the rock resistance (serial connection). The direct current (DC) flowing through of the rock is measured when excited with a reference voltage (i) at these electrode pairs (at the same depth) in order to gain an indication into the liquid water
content and (ii) between electrodes at different depth using and sensor-internal multiplexing unit. The latter configuration has to be interpreted carefully due to the extremely high resistances of this configuration (resistance measurements depend on the contact resistance of the electrodes and on local heterogeneity of the rock between these electrodes). While Table 3 provides the depths of the electrodes for each position, Figure 9 indicates a strong seasonal pattern, which is most likely related to the freezing of the rock. Comparable to the results of a study by Krautblatter (2009), temperature-resistivity gradients for intact
porous rock in frozen state here lie in a similar range of about $20-40\%/^\circ\mathrm{C}$ cooling (Hasler, 2011).

**Table 3.** Depths of thermistors and electrodes by position and medium under investigation.

| Position | Medium | Depths (cm) of thermistors[a] | Depths (cm) of electrodes |
|---|---|---|---|
| MH02 | Fracture | 10, 30, 70 | none |
| MH03 | Fracture | 10, 40, 60, 70, 75, 80, 85 | none |
| MH04 | Fracture | 5, 20, 30, 35, 40 | none |
| MH05 | Fracture | 10, 80, 150, 180 | 10, 80, 150, 180 |
| MH07 | Fracture | 10, 100, 200, 300 | 10, 100, 200, 300 |
| MH10 | Rock | 10, 35, 60, 85 | 9.5, 10.5, 34.5, 35.5, , 59.5, 60.5, 84.5, 85.5 |
| MH11 | Rock | 10, 35, 60, 85 \|[b] 5, 10, 20, 30, 50, 100 | 9.5, 10.5, 34.5, 35.5, , 59.5, 60.5, 84.5, 85.5 \|[b] none |
| MH12 | Rock | 10, 35, 60, 85 | 9.5, 10.5, 34.5, 35.5, , 59.5, 60.5, 84.5, 85.5 |
| MH27 | Rock | 5, 10, 20, 30, 50, 100 | none |
| MH30 | Rock | 5, 10, 20, 30, 50, 100 | none |
| MH47 | Rock | 5, 10, 20, 30, 50, 100 | none |
| MH46 | Rock | 5, 10, 20, 30, 50, 100 | none |

[a] Excluding single near-surface thermistors. [b] Intervention: change of sensor type.

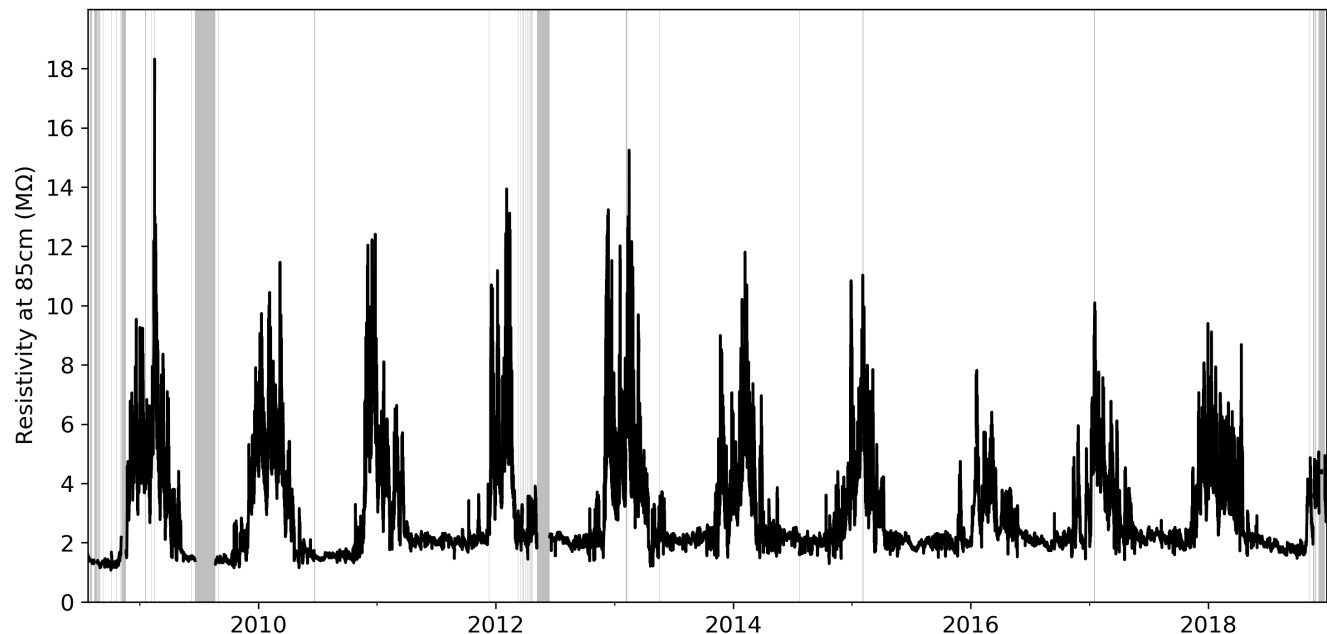

**Figure 9.** Resistivity time series measured at 85 cm depth in an intact rock wall at position MH10. Gray bars indicate data gaps.

### 4.4 Fracture displacements

Fracture displacements are measured using Stump/Terradata ForaPot crackmeters. These instruments are very accurate and robust linear potentiometers that are digitized using the wireless sensor nodes described earlier using a resistive half-bridge connection and a single-ended ADC channel per sensor element similar to the temperature measurements. The sensors exhibit a high linearity of $\pm 0.075\,\%$ (50-150mm measurement range) and $\pm 0.05\,\%$ (200-300mm measurement range) with a resolution better than $0.01\,\mathrm{mm}$ and a temperature dependant drift of max. $5\,\mathrm{ppm/^\circ C}$ i.e. $0.25\,\mathrm{\mu m/^\circ C}$ for a change of $10\,^\circ\mathrm{C}$ on a $50\,\mathrm{mm}$ range instrument. The devices are specified for operation in $-30$ to $100\,^\circ\mathrm{C}$. The setup has been validated on site with respect to device interchangeability and long-term stability, the details of which can be found in (Hasler et al., 2012) and the appendix of A. Hasler's PhD thesis (Hasler, 2011).

**Table 4.** Metadata describing all crackmeter sensors measuring fracture displacements, extended after (Weber et al., 2017).

| Position | 1st Crackmeter | 2nd Crackmeter | 3rd Crackmeter | Aspect | Slope | Fracture Characteristics |
|---|---|---|---|---|---|---|
| MH01 | 50 mm[a] | - | - | 95° N | 75° | intense solar radiation, microcracks next to main south facing fracture |
| MH02[b] | 50 mm | 150 mm[c]/−45° | - | 80° N | 50° | wet fracture system in main detachment zone, concave, often snowy |
| MH03 | 150 mm | - | - | 350° N | 65° | north-oriented, lower part ends in snow flank |
| MH04 | 50 mm | - | - | 320° N | 70° | debris ledge north of small saddle |
| MH06 | 100 mm | 200 mm/−90° | - | 90° N | 60° | south facing corner on ridge, often snowy |
| MH08 | 100 mm | 150 mm | - | 50° N | 90°/47° | wide, ventilated, shadowed main fracture |
| MH09 | 100 mm | 200 mm/54° | 200 mm/7° | 120° N | 65° | leaning tower buttress on top couloir exit |
| MH18 | 150 mm | - | - | 140° N | 20° | flat fracture, winter snow accumulation |
| MH20 | 150 mm | 150 mm/−60° | - | 70° N | 70° | bottom part of the fracture system in the main detachment zone , often snowy, wet fracture |
| MH21 | 100 mm | 200 mm[d]/−40° | - | 70° N | 85° | wide open, south exposed fracture on pillar below the detachment zone |
| MH22 | 100 mm | 150 mm/55° | - | 70° N | 85° | fracture system on ledge in north flank |

[a] Was removed for rock expansion test from 06/2010 to 12/2016. [b] The sensors were destroyed 2015-08-15 by rockfall. Crackmeters were re-equipped on 2016-07-28 but thermal measurements at this location was stopped. [c] Was 50 mm before 2016-07-28. [d] Was 150 mm before 2017-07-18.

The primary usage of these instruments is to determine displacements perpendicular to a fracture, i.e. the opening and closing movement (see Figure 2a). At select locations multiple crackmeters have been installed in order assess movement both perpendicular as well as parallel to the fracture (shearing) (see Figure 2b and c). In one location (position MH09) a triple crackmeter placement has been installed in order to capture three degrees of freedom of a large buttress detaching from the ridge into the East face. The buttress itself is additionally instrumented with a L1-GPS unit and integrated inclinometer (position MH35) mounted on top of the instable structure. Table 4 lists the details of all crackmeter installations: length of

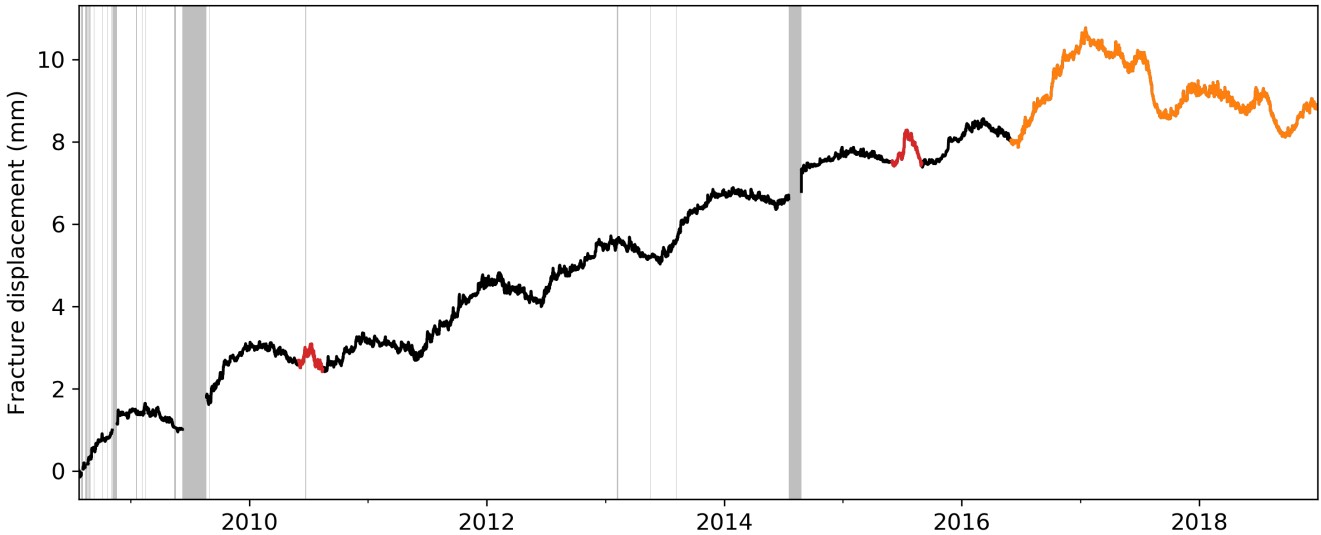

**Figure 10.** Fracture displacement measured perpendicular to the fracture at position MH03. Gray bars indicate data gaps.

each instrument, aspect, slope angle and characteristics. In cases where multiple crackmeters are mounted on a single location, the angle $\alpha$ between the two crackmeters (see Figure 2b) is given in combination with the length of the instrument. Using this information it is straightforward to calculate movement vectors in other angular configurations, e.g. parallel to the fracture using trigonometric equations (for details see Hasler et al., 2012; Weber et al., 2017). An example of the fracture displacement
measured perpendicular to the fracture at position MH03, a north-oriented fracture in a very thin segment of the ridge that remained after the July 2003 rockfall, is shown in Figure 10. The signal shows both cyclic behavior following the annual temperature regime as well as an irreversible component continuously widening the fracture. This figure is an example that although a seemingly regular behavior can be seen for many years (see black line) it is likely that further processes are involved. In this case, these processes led to additional small excursions in summer 2010 and 2015 (see red lines Fig. 10) as well as to
a change in the regime from ca. 2017 onwards (see orange lines Fig. 10) where the "regularity" of the preceding years is perturbed.

### 4.5  High-resolution visible light imaging

A time-lapse camera based on a Nikon D300 camera with a 24mm f/2.8D fixed focal length lens has been implemented using the PermaSense base station hardware and a WLAN data link (Keller et al., 2009b). The schedule and parameters for taking
pictures can be remotely managed, making it possible to control the camera based on experimental needs. At times when there are no imaging jobs active, the whole system sleeps minimizing overall power consumption to be woken up on request using our low-power wireless sensor network. In this manner, the camera has been operating since 2009, taking many tens of thousands of images from the field site. We have included a selection of images taken at approximately 2-hr intervals at full resolution of the camera (DX format sensor at $23.6\,\mathrm{mm} \times 15.8\,\mathrm{mm}$, $4288 \times 2848$ pixels, JPEG format). Further images

are available in the form of a hand-selected and labeled data set in (Meyer et al., 2018) or directly from the web frontend at http://data.permasense.ch where different resolutions and image formats are also available (select pictures in Nikon RAW (NEF) and/or in variable image resolution).

## 4.6 GNSS raw observation data

In order to assess large-scale movement of individual buttresses of the ridge a number of GNSS sensors are used. A high-performance Leica GRX1200+ GNSS receiver with a Leica AR10 antenna has been installed on the top outcrop of the lower ridge of the detachment scarp in December of 2010 (position MH42/HOGR). Low-cost wireless L1-GPS systems based on a u-blox LEA 6T receiver and a Trimble Bullet III antenna are mounted at further locations. Typically, this data is post-processed using double-differencing GPS processing along short baselines to derive daily position coordinates (see Section 5.2). The
position MH42/HOGR is acting as a reference. Since this constitutes a one-of-a-kind data set and other usages of this data are possible (Hurter et al., 2012) we are including the raw GNSS observations as well as the derived data products in this data set.

  Different GNSS observables are available depending on the receiver architecture used. The raw observables are available in the form of industry standard daily RINEX 2.11 observation files for each station concerned. Position MH42/HOGR contains both GPS and GLONASS observation data for both L1 and L2 sampled at an interval of $30\,\mathrm{s}$ while the remaining positions are
L1-GPS observations tracked at intervals of $30\,\mathrm{s}$ respectively $5\,\mathrm{s}$ (see Table 5).

**Table 5.** Details of GNSS observation periods and observables.

| Position | Period of Operation | Observables | Sampling Interval |
|---|---|---|---|
| MH42 / HOGR | 12/2010 - ongoing | C1 L1 D1 S1 P2 L2 D2 S2 | $30\,\mathrm{s}$ |
| MH33 | 08/2014 - ongoing | C1 L1 | $30\,\mathrm{s}$ |
| MH34 | 08/2014 - ongoing | C1 L1 | $30\,\mathrm{s}$ |
| MH35 | 06/2015 - ongoing | C1 L1 | $30\,\mathrm{s}$ |
| MH40 | 06/2015 - ongoing | C1 L1 | $5\,\mathrm{s}$ |
| MH43 | 08/2018 - ongoing | C1 L1 | $5\,\mathrm{s}$ |

## 4.7 Inclinometer data

The wireless L1-GPS sensor systems installed on positions 33, 34, 35 (stations MH33, MH34, MH35, see Table 5) also contain an integrated 2-axis inclinometer based on a MEMS component (Murata SCA830-D07). It is sampled every $120\,\mathrm{s}$, support a $\pm 30\,\mathrm{mg}$ offset accuracy over the operating temperature range. The data is transmitted over the wireless sensor network and can
be used to assess the rotational movement across the two horizontal axes of the rock mass as well as the height of the mast the GPS sensor is mounted on. For an example of this method see (Wirz et al., 2013, 2014) an example of the inclination change combined with displacement derived from daily GNSS position coordinates is shown in Figure 14 for position MH34.

## 4.8 Further data and related work

In the following, we list different data types and respective sources of data that we know exists and that is closely related to the data collated and documented in this publication and that are not available through a well established (national) data service e.g. weather service or cartographic service. It is a mixture of data that either we obtained by ourselves but is out of scope of this publication either (i) because it is specific to a campaign or purpose, (ii) not mature enough in the sense of quality control and processing or (iii) owned by a related (research) project effort. Nevertheless we take the opportunity to list the data sources we are currently aware of as of writing of this publication. For access to the respective data please contact the data owners given in the references.

### 4.8.1 Meteorological data

The closest comprehensive meteorological data record relative to the Matterhorn field site are the MeteoSwiss stations Stafel (VSSTA), Findelen (VSFIN), Gornergrat (GOR), Monte Rosa Plattje (MRP) and Zermatt (ZER), the MeteoGroup station Kleines Matterhorn as well as the stations of the Intercantonal Measurement and Information System (IMIS) ZER1, ZER2, ZER4 and GOR2. If required, these data have to be retrieved from the respective data owners.

### 4.8.2 Acoustic and microseismic data

Since 2012 a number of different experiments investigating acoustic emission (Weber et al., 2018c), microseismic signals (Weber et al., 2018b) using different instruments ranging from piezoacoustic sensors (>5 kHz), accelerometers (10 Hz-10 kHz) and seismometers (1-100 Hz) have been conducted. The respective data sets for these publications are publicly available and described in detail here (Weber et al., 2018a; Meyer et al., 2018). While the acoustic emission and mid-frequency accelerometer data is highly site specific and experimental, the lower frequency seismmometer data is of a more general interest and applicability. Since the end of 2018 this data is being propagated automatically to the Swiss Seismological Service (SED) at ETH Zurich where it is curated and can be accessed online.

Further seismic data originating in a measurement campaign of ARPA VDA, Italy from 2007 to 2012 near the J.A. Carrel hut on the south-east ridge of the Matterhorn at $3829\,\mathrm{m}$ a.s.l. is also available (Coviello et al., 2015; Occhiena et al., 2012).

### 4.8.3 Aerial imaging campaigns

In the year 2013 the UAV company senseFly in collaboration with Pix4D and Drone Adventures performed a demo flight with their UAV drones covering the whole Matterhorn from summit to base. From this campaign a 300 million points 3D pointcloud as well as orthophotos exists. Complementary imaging and scanning products are available by Swisstopo (www.swisstopo.admin.ch).

#### 4.8.4 Terrestrial laserscanning and radar campaigns

Several campaigns using terrestrial laserscanning (TLS) with instruments located both on the Matterhorn Hörnliridge and near the Hörnlihütte below (in 2014, 2015, 2016, 2018) as well as two real aperture radar interferometry (Caduff et al., 2015) campaigns (2015, 2016) have been performed. This data can be obtained from the authors upon request.

#### 4.8.5 Permafrost thermal data

A number of permafrost monitoring boreholes exists in the vicinity. The closest relative to our site are the PERMOS borehole Matterhorn (MAT_0205) (Luethi and Phillips, 2016; PERMOS Database 2019, 2019; Noetzli et al., 2019) located at the Hörnlihütte at $3270\,\text{m}$ a.s.l. and two shallow boreholes located at the J.A. Carrel hut on the Italian ridge (Coviello et al., 2015; Occhiena et al., 2012). Further downslope are the Cima Bianche field site managed by ARPA VDA and located on the Italian side (Pogliotti et al., 2015) at $3100\,\text{m}$ a.s.l and another borehole managed by SLF/Zermatt Bergbahnen and located on the Swiss side at Hirli near the ski lift station at $2775\,\text{m}$ a.s.l. Together with two GST temperature loggers located at the Matterhorn summit and operated by ARPA VDA this data constitutes a unique transect both with respect to the altitude profile but also the exposition. Up the south side, over the summit and down the northeast.

#### 4.8.6 Wireless network related technical data

A large amount of data concerning sensor status and health, network performance, solar power generation etc. is available over the whole deployment period. The PermaSense wireless sensor network on the Matterhorn constitutes the longest running sensor network for scientific (research) purposes worldwide and arguably also in an extreme environment. This data can be accessed through our online data portal at http://data.permasense.ch but publishing this data within this publication is out of scope.

## 5 Derived data products, processing and validation methodology

For a select amount of the primary data provided with this paper we present derived data products: A number of data sources exhibit very high sampling rates. Depending on the analysis goals these high sampling rates (e.g. $120\,\text{s}$) can be seen as an asset, e.g. to understand small scale, short term process chains but in general when dealing with the whole data set over a decade the gigantic amount of these data constitutes a burden. Therefore, we first introduce a method to downsample these data to reasonable rates in combination with a few data cleaning steps that have emerged as successful out of good practice. Specifically, this method includes (optional) filtering based on sensor-integrated reference resistors (for thermistors and crackmeters), data cleaning based on the manual interventions recorded and the temporal aggregation over 1-hour windows. The resulting data products are file sizes in the order of $100\,\text{kB}$ per year rather than 100's of MBs. We provide both a description of the method, the code implementation as well as all input and output data in the context of this paper to allow full transparency

and reproducibility. Furthermore by providing a toolset used for all processing steps concerned the reader can adapt processing steps or update the data set independently from future data set updates (living data process).

In the case of the GNSS data the raw GNSS observables are processed to daily positions using double-differential post processing and a local geodetic network as described in Section 5.2. A description of the processing toolset is available in Appendix A2.

## 5.1 Weather station, ground temperature, resistivity, fracture displacement and inclinometer derived data products

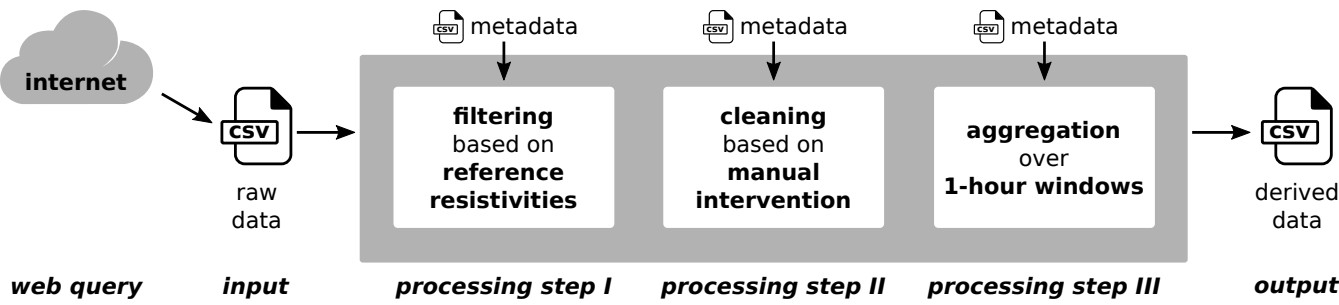

**Figure 11.** Three-step data processing methodology for PermaSense sensor data.

The data stored in the PermaSense GSN public database contains data obtained from sensor nodes after unit conversion. These data that, we call *raw data* can be downloaded using a standard web query (see Appendix B). However, since these data are sampled and transmitted independently they do not have a common time stamp and can at times contain discrepancies such as spurious outliers or the response to anthropogenic interventions, e.g. on manual service days. Therefore, a multi-step data processing methodology (see Figure 11) is applied, where each step is optional/user selectable (details are given in Appendix A1):

**Step I: filtering based on reference resistivity data**[1]  Independent additional electrical resistors are built into the PermaSense sensor chain, PermaSense sensor rod $12\,\mathrm{mm}$ and PermaSense sensor rod $20\,\mathrm{mm}$ as a means to assess sensor and data integrity (detailed description is given in Section 4.2 and 4.3). After filtering using these reference values, only data with reference resistivity values within a given range (defined in the metadata) are considered for further propagation.

**Step II: cleaning using a lookup-table**  Artifacts in the data either identified manually or systematically known (e.g. on device change interventions) are cleaned using this step. Cleaning operations are *delete*, *set an offset* or *replace* a single or multiple data points.

**Step III: aggregation over 1-hour windows**  For all data types but GNSS data and photographs 1-hour aggregates are calculated. For most data types, the aggregation function *arithmetic mean* was applied. Different aggregation functions were applied to some meteorological data, as an example *sum* for rain duration or *maximum* for rain peak intensity. For details, see Table A1 in Appendix A1.

## 5.2 GNSS derived data products

Daily static positions for all GNSS stations are calculated using double-differential GPS post processing based on two different tool chains: using the Bernese GNSS Software (Dach et al., 2015) and the open-source RTKLIB toolchain (Tomoji, 2018). For processing the observables are first collected from the online database and stored in daily observation files with one file per day and position. Double-differencing achieves best accuracy when utilizing the precision final GNSS data products from IGS although other GNSS data products can be used as well. In a final step the position coordinates are converted from WGS84 coordinates to Swiss national coordinates using the online REFRAME conversion service (REST API) by swisstopo. The resulting position data is subsequently uploaded again to the GSN database server from where it can be queried. The geodetic datum of all daily position data is CH1903+/LV95 with the reference frame Bessel (ellipsoidal). After post-processing data for a required amount of days, position data for each position is collated in a single file per position and a number of standardized graphs are generated (see Figure 12).

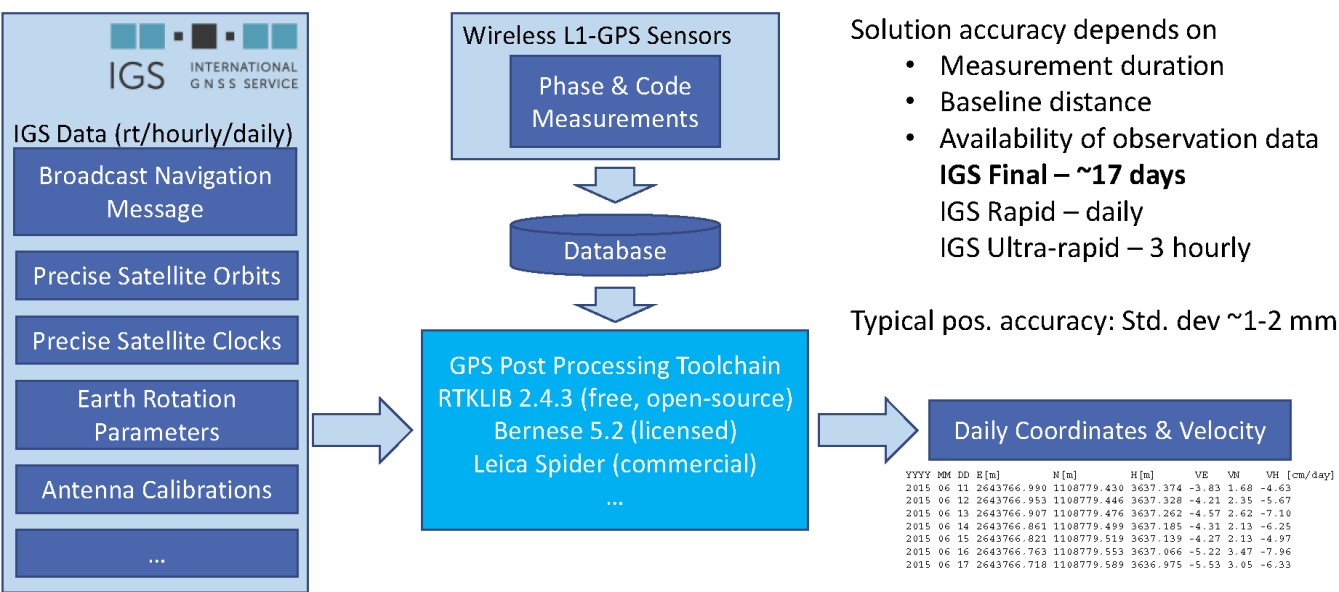

**Figure 12.** Differential GPS processing workflow.

Apart from the raw GNSS observations in the form of daily RINEX 2.11 files we provide the calculated daily positions for both processing toolchains as described above. Further, we provide the scrips and configuration files used to run the open-source RTKLIB toolchain both from prepared RINEX files and from the online data from our database (see Appendix A2). Double-differential GNSS processing (Teunissen and Montenbruck, 2017) is based on data obtained in a common observation interval from a station pair. Positions for the so-called "rover" can be calculated with high accuracy under the assumption that the "reference" station location is quasi-stationary and that observations from both stations are subject to similar perturbations. In practical application of this technique care should be taken that the baseline distance between any station pair is short, the

field of view to the satellites (horizon) is similar and that a station pair be located in the same altitude regime. Main quality indicators of the input data (GNSS observables) are the number of visible satellites, the signal-to-noise ratio and the observation duration. For the derived data products the ratio of fixed ambiguities as well as the standard deviations per coordinate axis are the key indicators.

## 5.3  Cross-validation of different sensor data: Examples

In this section we are giving a few select examples of data originating from different sensors plotted side-by-side in order to put this data into context. The few examples shown can by no means be exhaustive and are meant only as indicative examples to showcase some selected data in a visual format. We are only giving a brief introduction and interpretation in the following. Detailed analysis using further methods, especially by leveraging correlation methods that allow to combine data from different sensor types, should be applied to this data, but this is clearly beyond the scope of this paper.

In Figure 13 we showcase three types of data in a format suitable for the analysis of frozen ground: Fracture displacement measured using a crackmeter, rock internal resistivity and relative displacement measured using GNSS side-by-side and plotted against temperature, the different years are color coded in order to understand the behavior over time. The data shown originates from four different sensor types at three different locations. All three plots show freeze-thaw related processes that repeat each year as well as an irreversible kinematic component that dominates in summer when temperatures in the rock wall are well above zero.

Similarly, stepwise displacements can be seen when plotting GNSS derived daily positions and a co-located inclinometer on a conventional plot using time on the X-axis (see Figure 14). The first thing to note in this plot is the fact that different sensors and their resulting data types exhibit significantly different error patterns. Here, although the displacement is only on the order of millimeters, the GNSS derived displacements are much more accurate/stable than the inclinometer data that seem to be heavily influenced by present weather conditions, e.g. wind. Over winter periods, the displacement is negligible while the inclinometer raw data apparently relaxes. With the onset of the snow-melt period, an acceleration takes place that can be seen both in the GNSS data as well as the inclinometer. This acceleration continues until late fall. The exact timing of this behavior is known from in-depth analysis of the crackmeter data at Matterhorn (Hasler et al., 2012; Weber et al., 2017).

In the case of the GNSS positions at Matterhorn Hörnligrat all rover positions MH33-MH40, MH43 (the L1-GPS systems) are calculated relative to the two-frequency high-performance GNSS receiver located at MH42/HOGR. However this reference location is also exhibiting significant movement as it is positioned on the top of the buttress between the detachment zone in the second couloir and the first couloir. Therefore the absolute position output of positions MH33-MH40, MH43 contains the movement of the reference position MH42/HOGR. In order to quantify this movement and remove the differences from the rover positions MH33-MH40, MH43 precise absolute positions for MH42/HOGR are calculated using a longer baseline to the non-moving reference station of the Automated GNSS Network of Switzerland (AGNES) operated by swisstopo with station ZERM located at Furi, Zermatt, Switzerland, $1867\,\mathrm{m}$ a.s.l. The daily position data series provided with this paper contain the uncorrected position data. Calculating the corrected position values by differencing is straightforward. An example of such corrected data (the relative displacement of the positions MH33-35 and MH42/HOGR) can be seen in Figure 15.

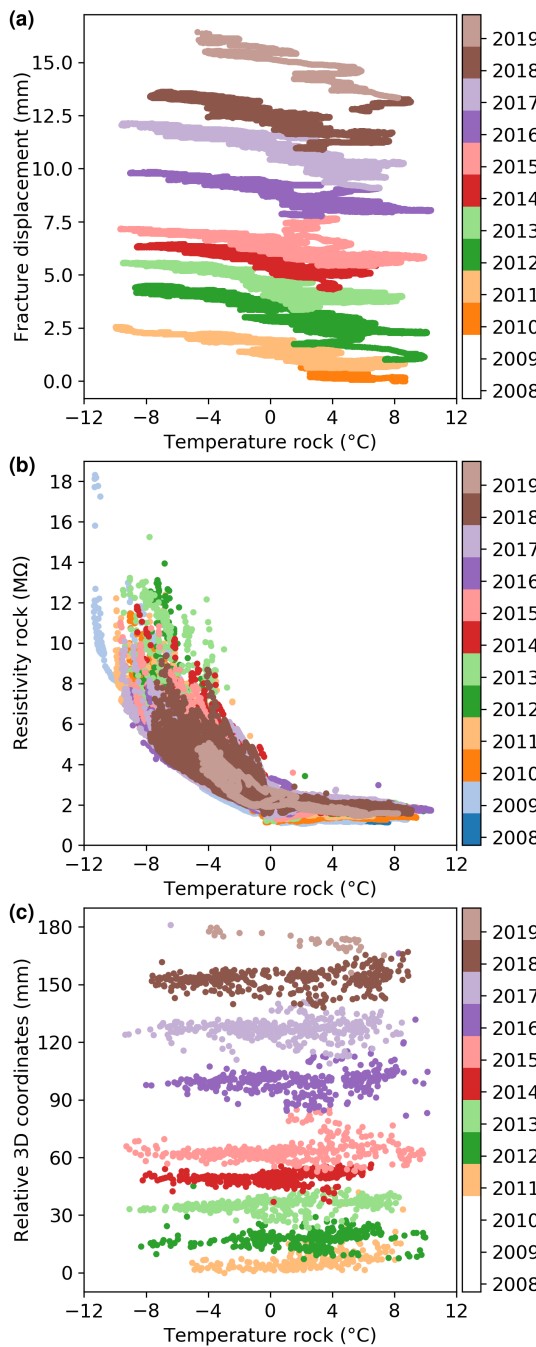

**Figure 13.** Displacement (MH08, dx2), resistivity (MH10, 85cm) and relative 3D coordinates derived from L1/L2 GNSS (MH42/HOGR) plotted against rock temperature (85cm) at position MH10.

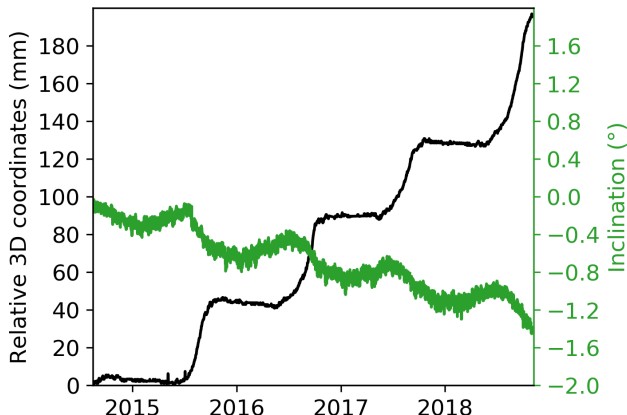

**Figure 14.** Displacement (black) and inclination (green) measured using an L1-GPS sensor for deriving displacement based on daily position data and an integrated 2-axis inclinometer sensor at position MH34 (the tower feature in the middle of Figure C10).

Similarly calculating velocities or aggregate displacements using a simple or more complex method (Wirz et al., 2014) is at the discretion of the data user.

## 6   Scientific results based on Matterhorn Hörnligrat data

Data over the period 2008-2011 were the foundation of A. Hasler's PhD thesis (Hasler, 2011) that investigated the thermal and
kinematic regime in steep bedrock permafrost for the first time to this extent and level of detail with important contributions to the spatial variability of the thermal regime (Hasler et al., 2011b) and kinematics (Hasler et al., 2012) concluding that enhanced movement in summer originates from hydro-thermally induced strength reduction in fractures containing perennial ice. This hypothesis was later supported when further data became available over a longer monitoring period (Weber et al., 2017). Further, in the wider context of rock slope stability assessment, a new metric was proposed to quantify irreversible
displacement of fractures based on the statistical separation of reversible components, caused by thermo-elastic strains, from irreversible components due to other processes (Weber et al., 2017). With the addition of acoustic emission and microseismic sensors to the field site S. Weber's PhD Thesis (Weber, 2018) focused more on structural aspects and the characterization of micro-seismic response to fracture events (Weber et al., 2018c) and on ambient vibrations (Weber et al., 2018b) with the following major findings: (1) A significant amplification of micro-seismic signals in the frequency band $33-67\,\mathrm{Hz}$ was found.
Filtering in this specific frequency band enables a more reliable detection of fracture events, which is a prerequisite for rock slope stability assessment and early warning. (2) The characterization of the site specific seismic response based on ambient seismic vibration recordings suggests that the temporal variations in resonance frequencies are linked to the formation and melt of ice-fill in bedrock fractures.

Along-side a number of technology-oriented publications have emerged that discuss sensor and wireless network design (Talzi
et al., 2007; Hasler et al., 2008; Beutel et al., 2009; Keller et al., 2009b; Buchli et al., 2012; Sutton et al., 2015b, a, 2017a),

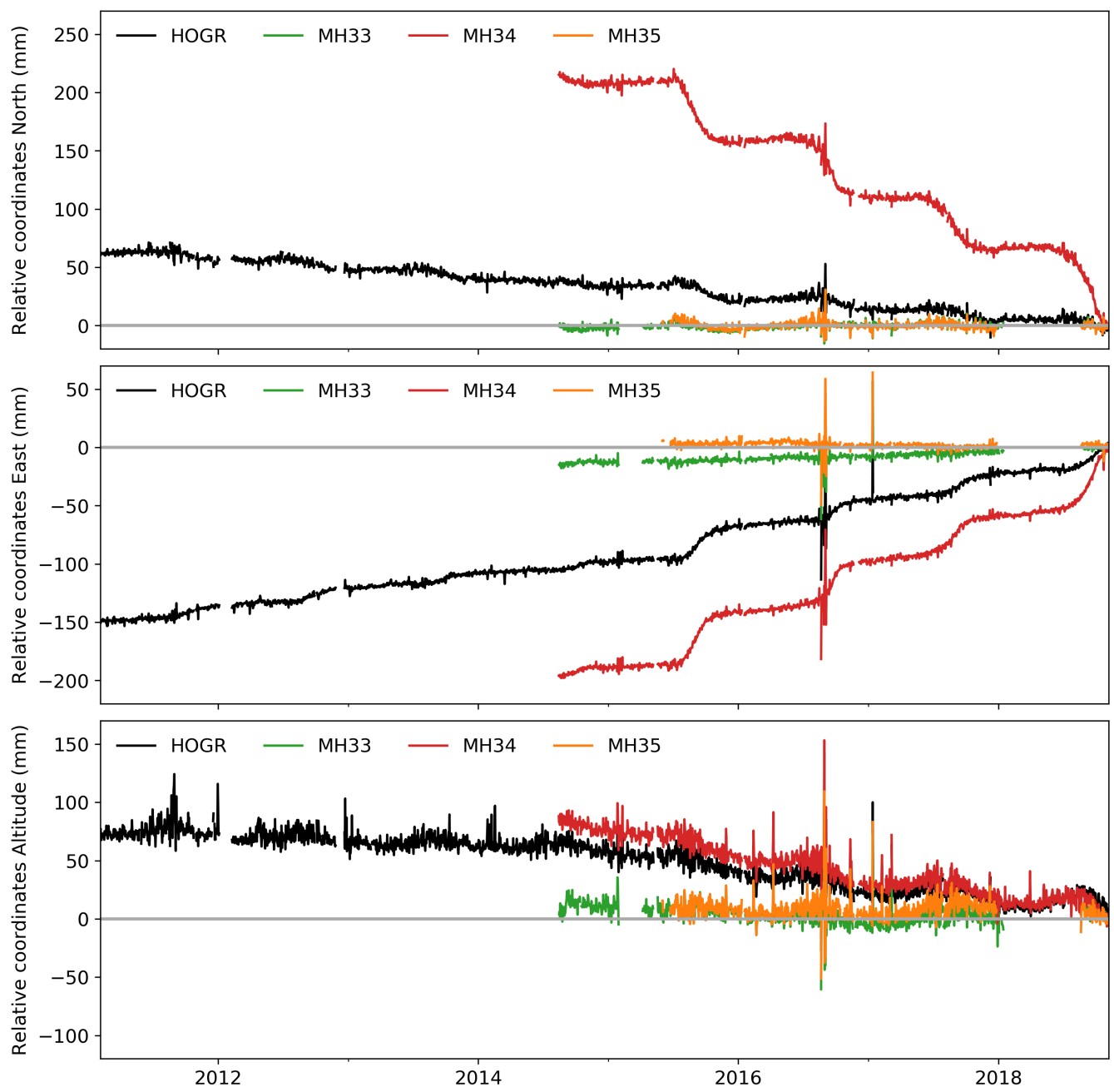

**Figure 15.** Relative displacement of the GNSS positions measured at Matterhorn Hörnligrat. For an approximate overview of the measurement setting see Figures C11 and C12.

performance analysis (Keller et al., 2012a, 2011) and smart sensors (Sutton et al., 2017b; Meyer et al., 2019a). More recently focus has shifted on even more complex sensing modalities including machine learning methods to the portfolio of application specific data analysis (Meyer et al., 2017, 2019b).

## 7 Code and data availability

The data set (doi.pangaea.de/10.1594/PANGAEA.897640, Weber et al., 2019) published with this paper contains data from the start of measurements on July 25, 2008 until December 31, 2018. An overview on the structure, file types and size of the data sets, both for the raw primary data and derived data products is given in Table 6. Furthermore the data set also contains the key metadata file for the Matterhorn field site: matterhorn_nodeposition.xslx. Annual updates of this data set are planned (living data process). Using the toolset described in Section 5 and using the online repository at http://data.permasense.ch (see Appendix A1 for details) the data user can also create custom updates of the data set independently.

**Table 6.** Structure, description, formats and sizes of the data set components.

| Directory | Data Description | Format | # Data Points | # Files | Size |
|---|---|---|---|---|---|
| gnss_data_raw | GNSS raw observations | RINEX 2.11 | 16'978'024 | 7'985 | 27.4 GB |
| gnss_derived_data_products | daily position data | csv | 7'578 | 48 | 243.2 MB |
| timelapse_images | time lapse images | jpg | 32'017 | 32'017 | 41.5 GB |
| timeseries_data_raw | raw primary sensor data | csv | 94'691'950 | 395 | 14.5 GB |
| timeseries_derived_data_products | sensor data after cleaning/aggregation | csv | 2'711'631 | 361 | 193.6 MB |
| timeseries_sanity_plots | standard plots for all data | png | - | 223 | 39.6 MB |
| matterhorn_nodepositions.xlsx | general metadata file | xlsx | - | 1 | 40 kB |
| README.md | - | md | - | 1 | 4 kB |
| Total | - | - | 114'421'200 | 41'031 | 83.8 GB |

The data sets as well as the toolset (code) for preparing, processing, validating and updating the data contained in this publication are available through the following providers and data links:

– Data set

https://doi.pangaea.de/10.1594/PANGAEA.897640, (Weber et al., 2019)

– Toolset (processing code)

https://doi.org/10.5281/zenodo.2542714, (Weber et al., 2019)

## 8 Conclusion and Outlook

When reflecting on the past ten+ years of development and operation it is fair to say that the promises of distributed wireless systems have delivered unprecedented detail and quality with respect to data. But on the other side the complexity and requirements for mastering increasing degrees of freedom increased as well. What has been especially troublesome at time was the sheer amount of data. Managing and especially the effort for devising a suitable data management system architecture including implementing workable and sustainable solutions has been greatly underestimated. There are no quick answers, make-or-buy decisions are frequently re-visited and there is no ready-made kitting that can be implemented swiftly. Since we believe that this present publication and it's related data set are already large and complex the acoustic emission and microseismic (AE/MS) data from Matterhorn as well as the terrestrial laserscanning and radar interferometry data are not included although it constitutes an integral part of the observations made at this field site. Parts of the AE/MS data has been published separately as we have indicated earlier, but putting all this into a single publication/data set would have simply been overwhelming.

The PermaSense data set from Matterhorn Hörnligrat is the largest, most fine-grained and diverse data set available for permafrost research worldwide. Remarkable about this data set is not only it's duration but also the diversity and density of measurements. The decade+ of interdisciplinary research summarized here shows in an exemplary way how modern (wireless) technological advancements enable new science and the related breakthroughs. The data described here is multi-facetted, exceptionally rich and therefore constitutes a substantial foundation for further research, e.g. in the area of methodology development, the development of process models, comparative studies, assessment of change in the environment, natural hazard warning and preparing for adaptation. Updates to the data set are planned (living data process) but independent of that the user can obtain updates independently using the toolset provided with this data set. Apart from flexibility, this allows also for maximum transparency and reproducibility of the data presented in this paper.

Opportunities for future work exist in a multitude of ways and we are only highlighting two directions here: Bringing together the data presented in this paper with data from our colleagues in Italy (ARPA VDA, Matterhorn summit, Carrel ridge, Cima Bianche monitoring site) and SLF/WSL + PERMOS (permafrost boreholes on the Swiss side) allows to obtain further detail over a large span of altitude regime of the European Alps as well as the peculiarities of north- vs. south-facing exposition. Comparative studies to other similar sites, e.g. Aiguille du Midi, Chamonix, France where the altitude and climatic forcing is similar but the morphology and especially the type of rock is very different are currently ongoing.

**Appendix A:  Toolsets for generating/processing the derived data products**

Code for the management and processing of data associated with this manuscript is available at https://doi.org/10.5281/zenodo.2542714, (Weber et al., 2019). It contains both a Python3 toolbox for downloading and processing primary data from the online web service at http://data.permasense.ch as well as scripts for post-processing GNSS data using the open-source tool RTKLIB (Tomoji, 2018). Detailed information how to run these tools is given in the README files therefore only a brief synopsis is presented here.

## A1    Filtering, cleaning and aggregation toolset

The GSN data management toolbox (Weber et al., 2019) is implemented in Python3. It allows to:

– Query data from PermaSense GSN server and save it locally as csv-files,

– Reload the locally stored csv-files,

– Filter according reference values if available,

– Clean data manually if needed,

– Generate 60-minute aggregates using in principle arithmetic mean (exceptions for weather data are shown in Table A1),

– Export yearly csv-files for each position/location,

– Generate standard plots for all positions/locations as sanity check and

– Query images from PermaSense GSN server and save it locally as jpg-files.

## A2    GNSS post-processing toolchain

The open-source RTKLIB toolchain (Tomoji, 2018) is a popular tool for processing GNSS data. It consists of a number of binary tools that can be used both in cmd-line mode and in combination with a GUI as well as the respective configuration files. In order to automate the processing of larger data sets we have developed a small toolchain that allows to prepare all data necessary and calculate double-differencing daily position solutions. In order to use this toolchain an operational installation of RTKLIB is required. For details on RTKLIB please refer to the respective tool documentation. The top-level shell script `compute_solution.sh` allows to specify a configuration parameter file, several options and the day for which processing is to be performed:

```
# Usage:
# compute_solution.sh -p [parameter-file] [-d] [-b] [-r] [-c] [-f] [-u] YYYY MM DD
#
# options:
```

**Table A1.** Aggregation functions used for the meteorological data at position MH25.

| Variable name | Aggregation function |
| --- | --- |
| rain_accumulation | sum |
| rain_duration | sum |
| rain_intensity | mean |
| rain_peak_intensity | max |
| hail_accumulation | sum |
| hail_duration | sum |
| hail_intensity | mean |
| hail_peak_intensity | mean |
| wind_direction_minimum | min |
| wind_direction_average | mean |
| wind_direction_maximum | max |
| wind_speed_minimum | min |
| wind_speed_average | mean |
| wind_speed_maximum | max |
| temp_air | mean |
| temp_internal | mean |
| relative_humidity | mean |
| air_pressure | mean |

```
#            -d: IGS data download
#            -b: no data download and no conversion for the basestation
#            -r: no data download and no conversion for the roverstation
#      -c: no conversion
#      -f: use IGS final data product
#      -u: upload to GSN database
```

The parameter file specified contains information on the baseline pair being processed, data products used and the exact locations of servers and directories to be used. The latter of which need to be adapted to suit your specific installation. The `compute_solution.sh` shell script calls further auxiliary programs written in python as well as tools from RTKLIB. The syntax is best explained using an example for computing positions MH42/HOGR and MH33 for the first day of the year 2017:

```
./compute_solution.sh -p parameter_file_HOGR_ZERM.txt -b -r -c -d -f 2017 01 01
./compute_solution.sh -p parameter_file_MH33.txt -b -f 2017 01 01
```

5    An example of how this toolchain can be used to compute daily positions for all Matterhorn GNSS positions for a given day is shown in the shell script `gps_batch_compute.sh` that can also be used to automate this process on a compute server.

**Appendix B: PermaSense online data access**

For use cases where updates to the data being provided with this paper or direct access to the online database is required we include a short introduction to the web interface and it's query syntax here. The data in the GSN database available at http://data.permasense.ch is organized in data structures called virtual sensors (VS) per deployment (see Figure B1). If there are multiple sensors yielding the same data types, this data is multiplexed into the same VS. Each VS has a unique name: `<deployment>\_<sensor type>`. For convenient data download the web frontend supports complex queries using the *multidata* query interface [2] of GSN with the following options:

- Data selection per field/VS

- Multiple output formats (xml, csv, images)

- Limits on the result set

- Aggregation of fields

- Conditions on fields

An example for a simple one-shot query without aggregation or further conditions for obtaining all fields of the matterhorn_displacement virtual sensor between 25/08/2012 and 13/06/2013 (UTC) is http://data.permasense.ch/multidata?vs[0] =matterhorn_displacement&time_format=iso&field[0]=All&from=25/08/2012+00:00:00&to=13/06/2013+00:00:00. Here `vs[0]` specifies the name of the virtual sensor, `time_format` specifies the time format of the returned data, `field[0]` specifies the list of data fields to return and the `from`, `to` clause limits the time window of the query. The result of this query is a CSV-formatted file with the requested data, in this case all sensor positions will be reported that produced data in the given time interval. Typically a query for data pertaining to a single position only will employ further limits, e.g. on the field position as follows for a limit to position 3: `c_field[1]=position&c_min[1]=2&c_max[1]=3`. The most relevant GSN multidata query syntax are given in Table B1.

---

[2]https://github.com/LSIR/gsn/wiki/Web-Interface

**Table B1.** GSN multidata query interface syntax.

**General Options**

| Option | Description | Allowed Values | Default |
|---|---|---|---|
| vs[n] | Virtual sensor, n specifies the number of the VS referenced in later options, e.g. vs[0]=ts, vs[1]=rh | All or the name of the VS | mandatory |
| field[n] | Parameter name | All or list of parameters | mandatory |
| time_format | The format of the time stamp | unix, iso | unix |
| download_format | The format of the download | csv, xml, pdf, jpg, nef | csv |

**Limits**

| Option | Description | Allowed Values | Default |
|---|---|---|---|
| nb | Enable (SPECIFIED) or Disable the count based limit | SPECIFIED, ALL | ALL |
| nb_value | The number of points (used where nb=SPECIFIED) | Number of points | none |
| from | Start time of the query | dd/MM/yyyy+hh:mm:ss | none |
| to | End time of the query | dd/MM/yyyy+hh:mm:ss | none |

**Aggregation**

| Option | Description | Allowed Values | Default |
|---|---|---|---|
| agg_function | The aggregation function applied to the data | avg, max, min, -1 = disabled | disabled |
| agg_period | The period over which to aggregate | Value, -1 = disabled | disabled |
| agg_unit | A multiplier for the aggregation period | 1 = ms, -1 = disabled | disabled |

**Conditions**

| Option | Description | Allowed Values | Default |
|---|---|---|---|
| c_join[n] | Logical conditions for complex clauses. | and, or, -1 = disabled | disabled |
| c_vs[n] | The virtual sensor to which the condition is to be applied. | All or the name of the vs | All |
| c_field[n] | The parameter to which the condition is to be applied. | All or the name of the parameter | All |
| c_min[n] | The minimum value of the condition to be met. | -inf or the minimum value | All |
| c_max[n] | The maximum value of the condition to be met. | -inf or the maximum value | All |

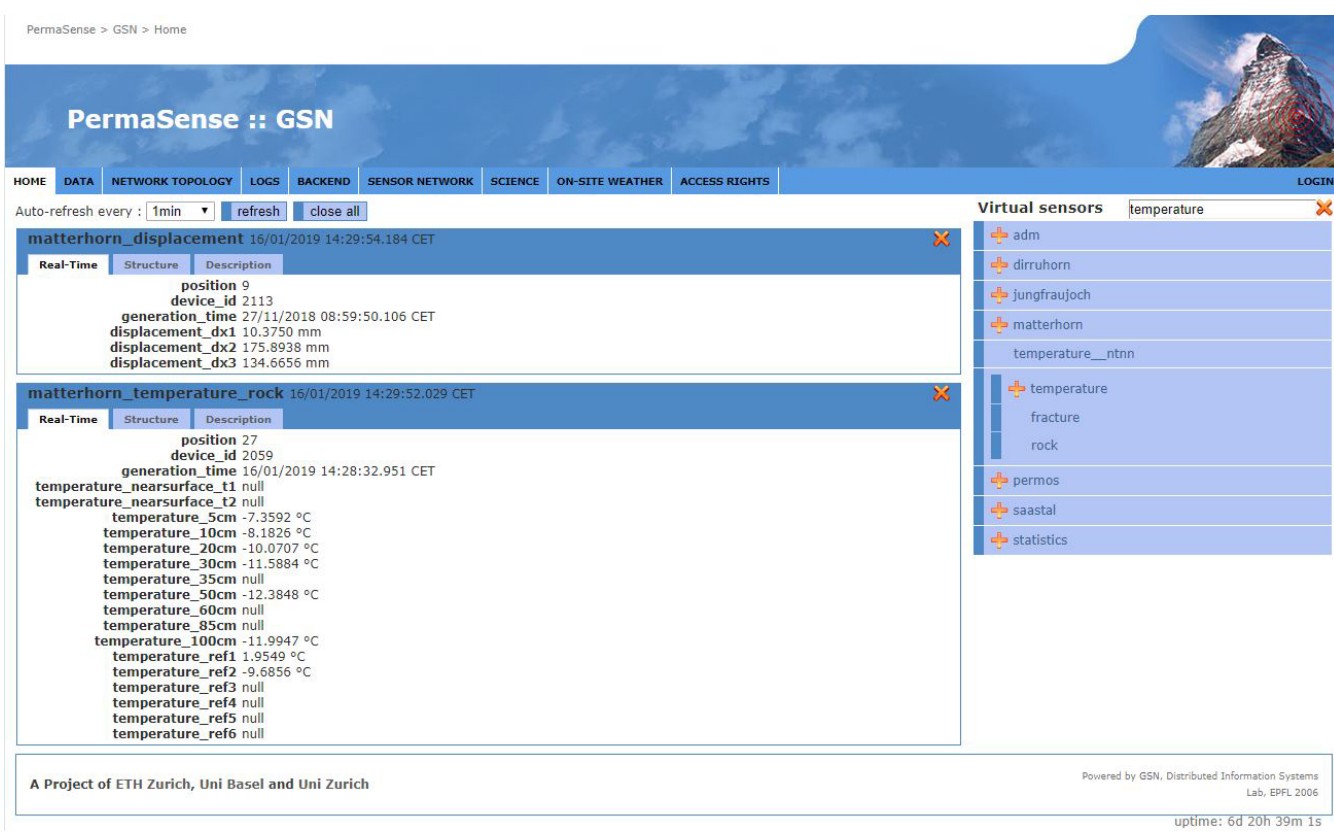

**Figure B1.** The online data management web frontend at http://data.permasense.ch allows to access all data in real-time. Data are accessed by data type in entities called virtual sensors (right). Selected standard views, e.g. key graphs can be accessed via the tabs at the top.

**Appendix C: Pictures of the field site and selected instrument details**

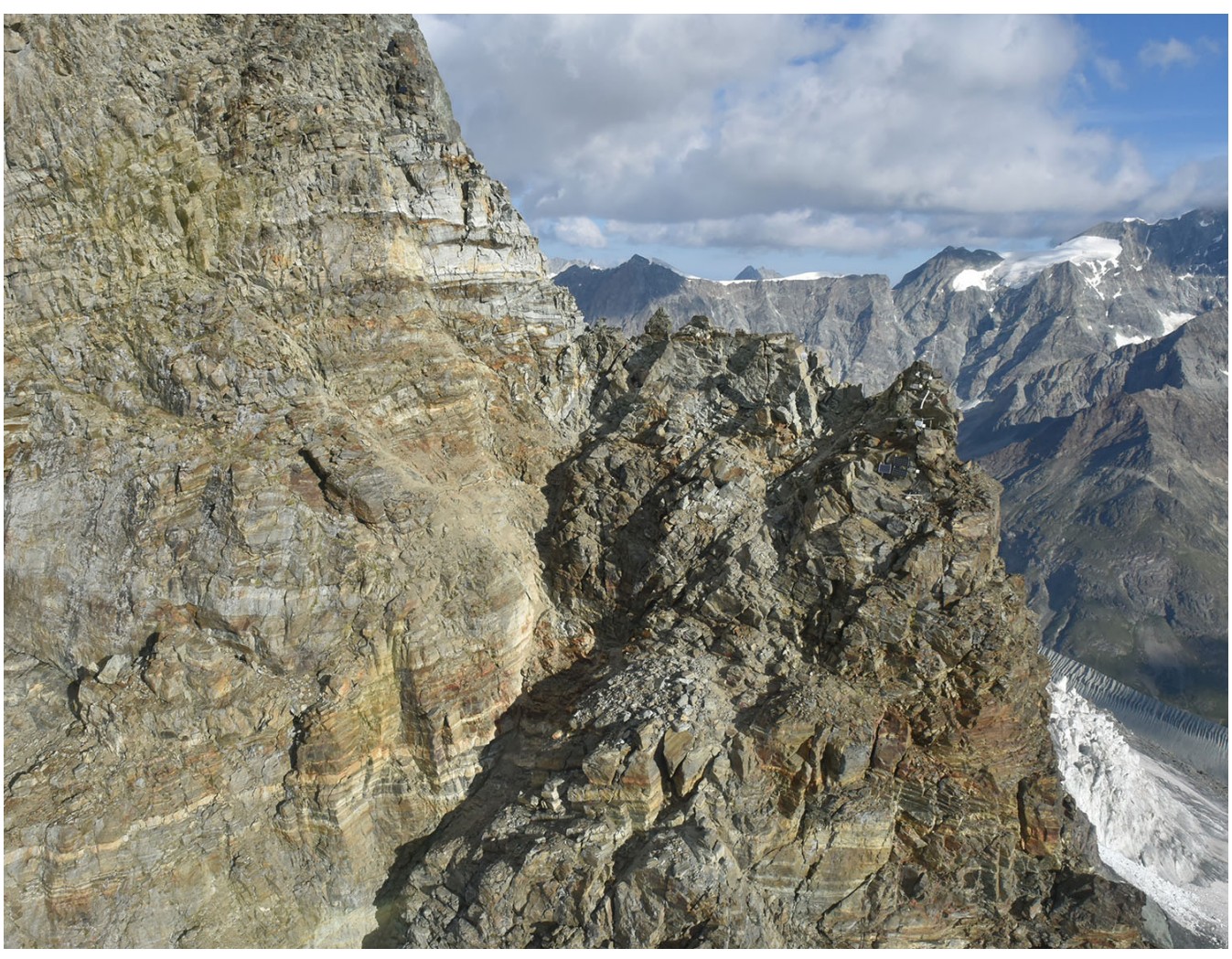

**Figure C1.** From the south the large detachment scar (light grey rock) to the left of the deeply incised second couloir on Matterhorn Hörnligrat is well visible.

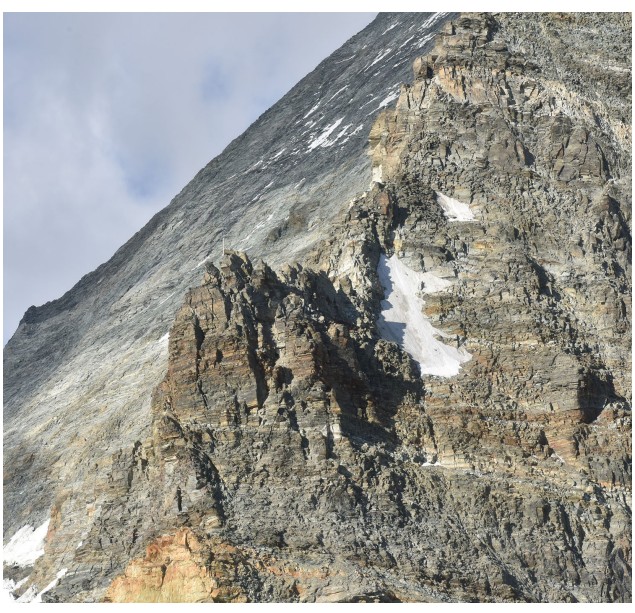

**Figure C2.** North of the detachment zone (light grey colored rock) a small ice field is visible delimiting the strongly fractured topography close to the ridge from the north face.

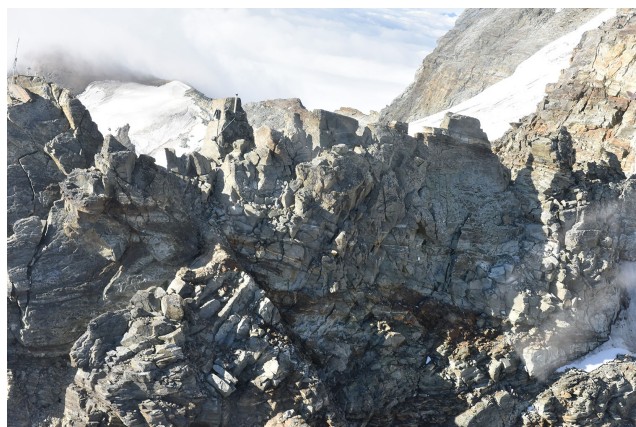

**Figure C3.** Close up from the north onto the Hörnligrat with the weather station visible on the top left and the MH11.

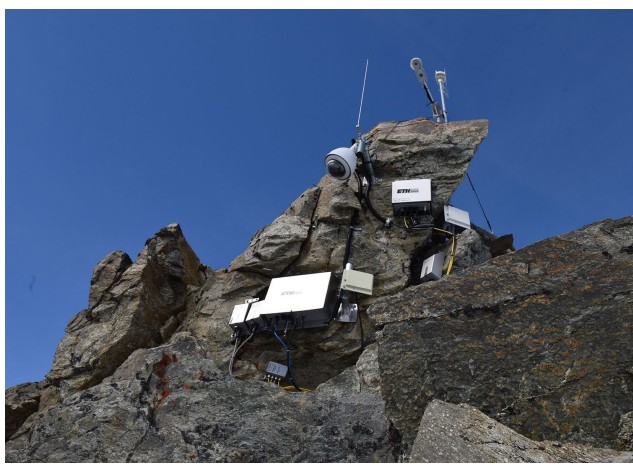

**Figure C4.** A Vaisala WXT520 weather station and Kipp & Zonen CNR4 radiometer are installed on top of the ridge crest. Other equipment shown here are a webcam, Leica GRX1200+ high-precision GNSS receiver and the required wireless transmission and power control equipment.

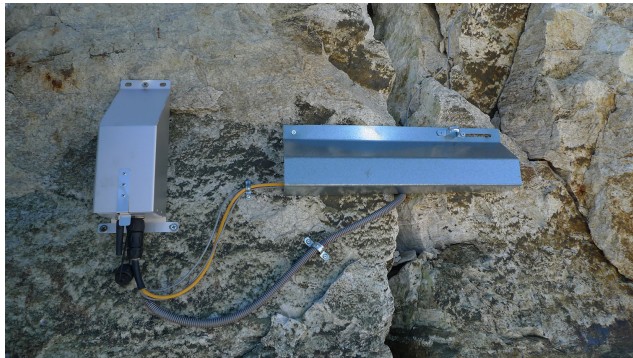

**Figure C5.** Close up of crackmeter and thermistor chain installation at position MH03. The wireless sensor node is housed in the steel protective shoe on the left while the crackmeter is located under the steel protective shield in the middle.

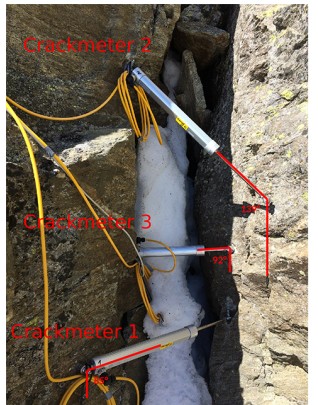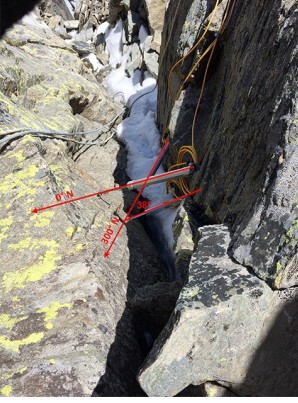

**Figure C6.** Sensor setup at position MH09 with 3 crackmeters and one surface thermistor channel.

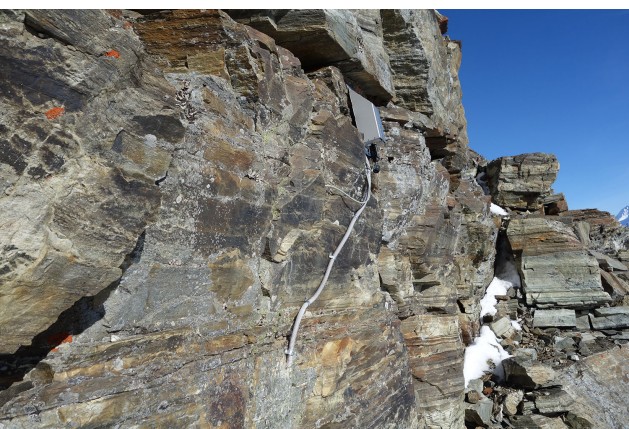

**Figure C7.** Rock temperature measurement at position MH10 on a south exposed rock face.

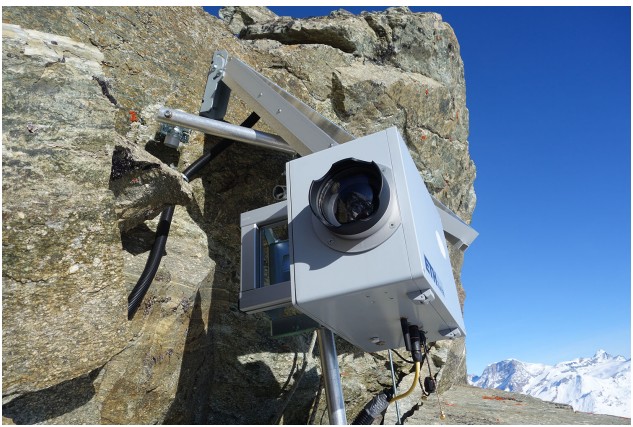

**Figure C8.** High-resolution time-lapse camera located at position MH19.

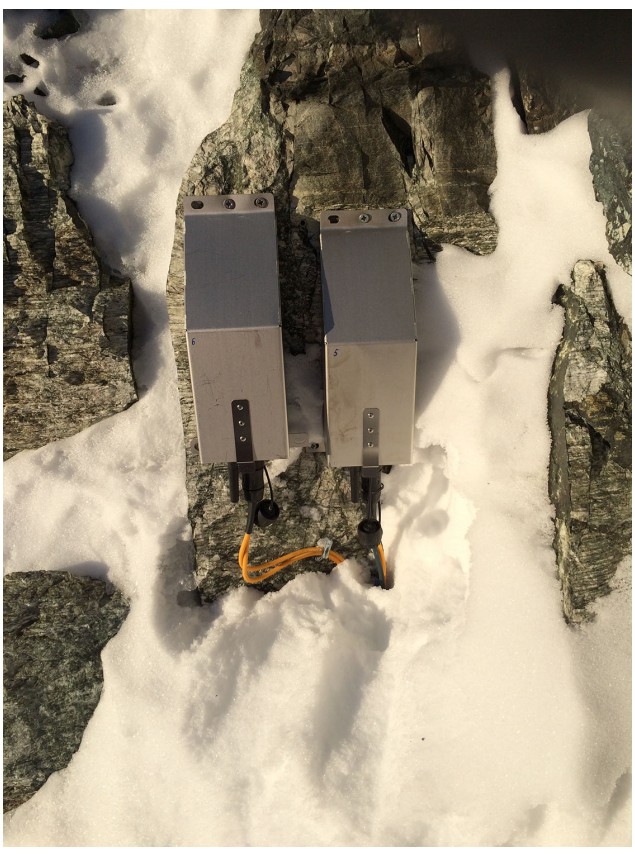

**Figure C9.** Sensor nodes at position MH05 and MH06 are installed on a small rock wall above a ledge to prevent heavy snow coverage while the sensors themselves are not visible.

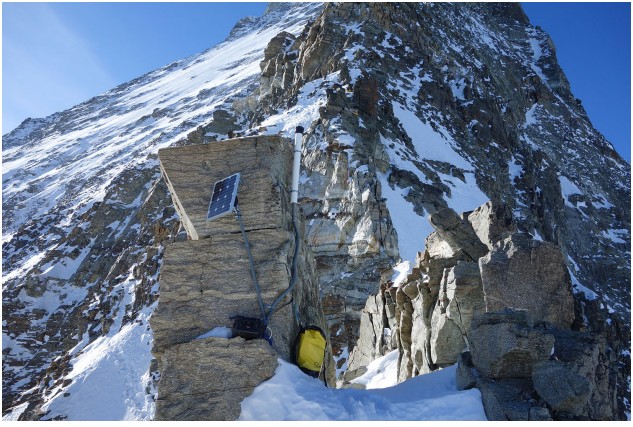

**Figure C10.** Wireless L1-GPS installed at position MH34 monitoring the gradual tilting of a little tower feature that is separated from the main ridge.

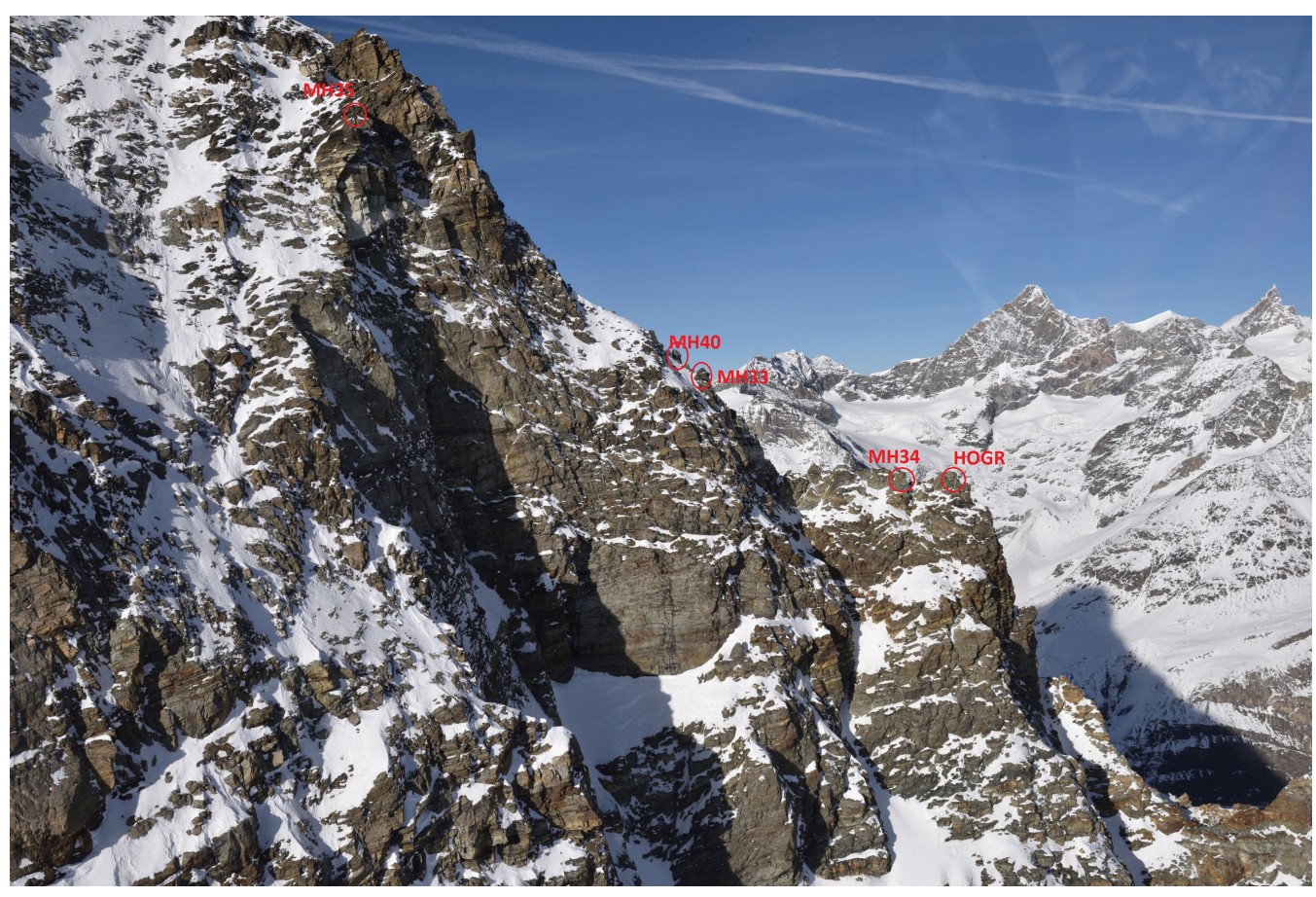

**Figure C11.** Winter view of the whole field site when approaching from the south. The red circles denote the GNSS measurement positions. The detachment zone is located in the shadow between positions MH33 and MH34.

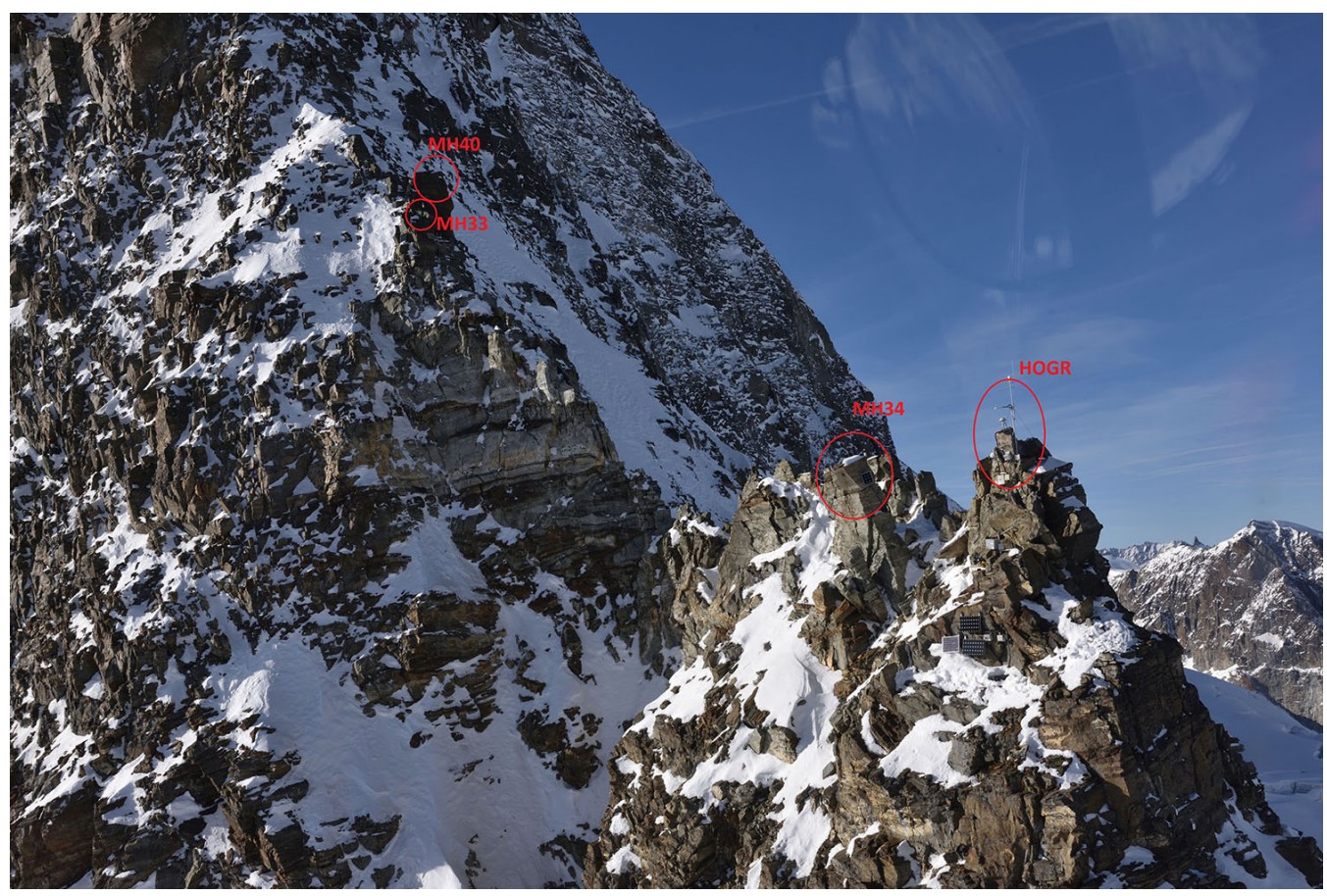

**Figure C12.** Closeup of the detachment zone (middle) and the buttress between first and second couloir. The instrument cluster around and below GNSS measurement position MH42 / HOGR (right) contains the weather station, webcam, high-resolution camera and sensor network base station.

*Author contributions.* Samuel Weber (SW) and Jan Beutel (JB) prepared the data set and wrote the manuscript with the help of all other authors. The processing code was developed together with Matthias Meyer (MM) and Tonio Gsell (TG). The original PermaSense project was conceived and implemented by Stephan Gruber (SG), Andreas Hasler (AH), Igor Talzi (IT), Christian Tschudin (CT) and Daniel Vonder Mühll (DV) with help from JB. The revised sensor system architecture was designed by JB, Matthias Keller (MK), Roman Lim (RL), Reto Da Forno (RD), TG, Mustafa Yücel (MY) and Lothar Thiele (LT). Alain Geiger (AG) and Philippe Limpach (PL) contributed to the GNSS sensors systems. The data management system was designed by JB, TG, RL, SG, AH, MK, MY and SW, operation was managed by JB, TG, SG, AH, SW, MK, MM and Andreas Vieli (AV).

*Competing interests.* The authors declare that they have no conflict of interest.

*Acknowledgements.* The early parts of this work have been funded through the Swiss National Science Foundation National Competence Center on Mobile Information and Communication Systems (NCCR-MICS), the ETH Zurich Competence Center on Environment and Sustainability (CCES) as well as the Swiss Federal Office of the Environment (FOEN) and later through nano-tera.ch, ref. no. 530659. Support in the form of equipment has been given by Hilti Schweiz AG, Arc'teryx and Beal. The technical workshops at ETHZ and UniZH as well as Art of Technology, Zurich contributed to the successful development and implementation of various pieces of equipment. We are indebted to the extraordinary local support we have received for our research activities in Zermatt: specifically the municipality of Zermatt, the whole team of Air Zermatt, Kurt Lauber, Stephanie Mayor and the Hörnlihütte team, Alpin Center Zermatt, Kurt Guntli (Zermatter Bergbahnen), Willy Gitz (Sprengtechnik-GFS), Hotel Bahnhof and Serac (Zermatt) as well as the local mountain guides Hermann Biner, Robert Andenmatten, Willy Taugwalder, Urs Lerjen, Benedikt Perren, Bruno Jelk, Hannes Walser, Simon Anthamatten and Anjan Truffer. Without this strong positive welcome this work would not have been possible. Thank you for the trust, confidence and help bestowed by the Swiss Federal authorities (Hugo Raetzo, FOEN) and Ct. Valais (Charly Wuilloud, Raphael Mayoraz) into our very special activities on a very special mountain. Martin Vetterli (EPFL) continuously motivated to push hard and think big, Hansueli Gubler (AlpuG, Davos) gave invaluable technical guidance on the first sensor concept, Karl Aberer, Ali Salehi and Sofiane Sarni (EPFL) helped with understanding the GSN data management system, Stephanie Gubler and Nicolas Dawes helped to structure our data, Elmar Brockmann (swisstopo, Wabern) was a regular source of GNSS related answers, Daniel Reinstadler (evoNET, Landeck, Austria) had the perfect solution at hand for transmitting large amounts of data to the Internet that we realized with the help of Maurizio Savina and Paolo Burlando (ETHZ), David Amitrano (Universite Joseph Fourier, Grenoble, France) helped to kick off our work on AE and finally, Philip Deline and Ludovic Ravanel (Universite de Savoie, Chambery) who were courageous enough to trust our technology in order to collaboratively equip Aiguille du Midi (Chamonix, France) with wireless sensors in a similar fashion. Many friends and helpers were involved in field work: Lucas Girard, Lorenz Böckli, Joel Fiddes, Luzia Fischer, Karl Stranski, Vanessa Wirz, Christoph Walser, Raphael Eiter as well as Marcia Phillips, Robert Kenner, Johann Müller (all three TLS) and our colleagues from the "other side" of the mountain: Paolo Pogliotti, Umberto Morra di Cella and Edoardo Cremonese (ARPA VDA, Italy). Last but not least we owe sincere thanks and gratitude to Jeannette Nötzli for tireless consulting concerning permafrost related issues, long-term monitoring strategy, data curation, policy and strategic decisions as well as to Wilfried Häberli for persistent guidance, judicious stewardship and friendship through all the years. Reviews from Paolo Pogliotti, and an anonymous referee provided valuable comments that helped to improve the paper substantially. Finally, we thank the handling Editor Kirsten Elger for constructive feedback and suggestions.

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
