# Peer review of "A decade of detailed observations (2008–2018) in steep bedrock permafrost at Matterhorn Hörnligrat (Zermatt, CH)"

_Earth System Science Data, 2019_

## Referee Comment (RC1) · Paolo Pogliotti (Referee) · 3 Apr 2019

**GENERAL COMMENT**

This manuscript presents an impressive dataset recorded over ten years on the north east ridge of Matterhorn, in steep bedrock permafrost conditions. The dataset includes multi-spot measures of rock temperature and resistivity at different depths, fracture kinematics, displacement and tilting of rock buttresses, time-lapse images and in situ meteorological data.
Data collection methods and sensors setup are exhaustively discussed. The dataset, that includes raw data and post-processed products, is openly accessible and available from a long-term data repository (Pangaea).
The dataset is new and extremely relevant for researchers studying the instability processes related to permafrost degradation in high-mountain environment. The variety of measures represents a great added value that allows the use of these data for developing, calibrating or validating process-oriented models as well as designing and testing of remote early-warning systems. Moreover a number of specific fields of research like e.g. geomatics, hydrology, datascience and more could potentially benefit of this dataset.

I have few **major comments**:

**1.** The manuscript provides all the informations needed to other researchers for an effective use of the dataset but sometimes these informations are lost in the complexity of the paper. This happens because the article is not exclusively focused on the dataset published in Pangaea but also on the Permasense web portal. Although the two objects are intimately linked, I believe it is important to keep them separate to promote the use of published data (that can be cited by others) rather than the use of real-time ones (that cannot). I strongly suggest moving all content related to access and use of the Permasense web portal, included the processing code published on zenodo, to appendix A or publishing it in a separate article.

**2.** The Chapter 3 – Technology is out of the purpose of this paper because of tailored on the hardware and software architecture of the GSN and the related storage infrastructure. These aspects are very important and impressive but, in fact, irrelevant for the effective use of the dataset. Also for this, I strongly suggest moving this chapter in appendix A or use it as core for a separate publication in another journal and keep in this publication just the essential (like e.g. figures 2 and 3).

**3.** In the context of environmental science and notably in permafrost studies, the site description (chapter 2) is important and should provide the reader (who does not know the area) with all the basic information, to frame the area in therms of climate, topography, morphology and geology.
In my opinion these general informations are totally missing in the manuscript and this gap must be filled. I strongly suggest to provide the following general informations:
- climate: temperature, precipitation, seasonal extremes, etc… on average
- topography/morphology: mean orientation of the ridge, mean aspect and slopes of the ridge flanks, elevation interval, etc...
- geology: main lithologies and rough structural setting (e.g. orientation of the main faults and fractures families…)
I guess that in Hasler's papers and PhD thesis all these infos are more or less ready to go.
See also the technical corrections for further comments.

4. Downloading the dataset from Pangaea repository I noticed that all the .csv files contains a lot of decimal places. In my opinion the dimension of the archives could be significantly reduced by rounding all the double values to the $2^{nd}$ decimal place. Please consider this.

**To conclude**, I believe the article is excellent for publication in ESSD after a careful reorganization of the contents to make it much leaner and exclusively focused on the Pangea dataset.

**Technical corrections & typing errors**

P2/R19-21: remake this sentence the sense is understandable but not easy readable. Maybe the brackets are not well positioned?

P2/R24: put a comma after "setting"

P2/R25: "data: The longest" uppercase after the colon. This is systematic in your paper but Mr. Google told me that is not usual: *'Capitalization: First Word After a Colon. In British English, the first letter after a colon is capitalized only if it's a proper noun or an acronym; in American English, the first word after a colon is sometimes capitalized if it begins a complete sentence'*.

P2/R25: remove the acronym w.r.t. whole over the paper (lot of occurrences), use the full words instead

P2/R31-32: this sentence is a kind of repetition, can be removed

P3/R9: what do you mean with 'sensor (type) extension'?

P3/R21-32: move&merge this block of text in the "research interest"-paragraph 2.1.
See also major comments n.3

P4/R4: please provide a most statistically significant MAAT, at least over the period 2008-2018. If possible provide also the mean annual precipitation (mm), seasonal extremes and other useful climatic informations. See also major comments n.3

P5/R22-25: 'All the … context', already said at the end of the Chapter 1-Introduction… remove or merge there.

P6-P11: See major comments n.1&2

P12/R16-18: a description of section 5 is not pertinent here. Move it to the end of the chapter 1 or delete.

P12/R19: It would be very useful to have the figure 16 in this chapter, between table 2 and figure 6. This is important especially for the comprehension of caption in table 2 and remarks in rows 19-26.

P12/R20: 'reliability' in place of 'availability'

P12/R23-24: 'Therefore… paper' is not necessary. This remark already rise up from previous sentences.

P12/R28-32: . Remove the first sentence that is well-known and move this block at the end of paragraph. That is, start the paragraph with 'Since 2010...'

P13/R1: in Table 1 WXT520/536 is cited, while in text just WXT520. In fact WXT536 appears just in tab.1. Please fix this inconsistency or explain it.

P15/R7: define 'near surface' in therms of depth… 10cm?

P15/R8: explain the acronym NTC

P15/R12: please, add a reference to Tab.1 after 'used'

P15/R12: remove the last sentence, it is a repetition of row P15/R8-9

P16/Fig.7: this figure is not cited in text (notably paragraph 4.1). Anyway I guess it is not very relevant, I let you decide whether to keep it or take it off.

P16/R4: explain acronym ADC. Please add the drift values also for this sensor (if possible)

P16/R10: table 2 shows the depths of temperature measures in rock face and fractures but the indication of which of the 4 thermistor systems is used is missing.
To address this task you can label (as e.g. A,B,C,D) the ground temperature in Tab.1 (first column) and text in this paragraph (P16/R1-9), then call the same labels in the caption of Tab.2 like e.g.: [a]Intervention: Change of thermistor system from A to D. [b]Intervention: Change of thermistor system from B to C. etc... Tab.2 might become little more complex but more exhaustive and meaningful.

P17/Fig.8 and 9: due to the length of the time series, evaluate of enlarging the plots exploiting the entire width of the page while keeping fix the present height.

P19/Fig.10: same as above

P19/R2: bis → is. This sentence could be replicated in the table's caption.

P19/R4: remove 'here'

P19/R6: add a comma after 'rockfall'

P19/R13: remove the last 'remotely'

P19/R14: comma after 'active'

P20/R2-4: please, indicate clearly in the first sentence that what you call the 'primary' antenna, works as a master station with respect to the others and that it is located at point MH42/HOGR. At first glance, it was not easy to be sure of that. I know this will be discussed further in section 5.2, but it is better to have it here too.

P22/R17-18: check the English, the sentence sounds strange

P22/R22-23: this sentence can be omitted.

P30/R24: use acoustic emission / microseismic  instead of AE/MS

P31/R1: See major comments n.1&2.

P32/TableA1: the reference to this table is missing in the text.

---

## Referee Comment (RC2) · Anonymous Referee #2 · 17 Apr 2019

Journal: ESSD Title: A decade of detailed observations (2008?2018) in steep bedrock permafrost at Matterhorn Hornligrat (Zermatt, CH) Author(s): Samuel Weber et al. MS No.: essd-2019-14 MS Type: Research article

Dear Editors,

This is very excellent manuscript since it showcases comprehensive and state-of-the-arts data sets recording thermal, hydraulic and kinematic dynamics of permafrost underlying alpine bedrock cliffs where few observations have been conducted in the world due to its harsh environments. Manuscript is also written very well with perfect information of technological and geographical settings, and data quality, accordingly I strongly

recommend it to be published ESSD after minor revisions.

There are two points that I would like to comment for further revisions. 1. '6. Example of data use' : In this section, the authors only listed some publications (phD thesis, etc., ) based on this data sets. I recommend to describe here more scientifically, e.g., what was new scientific findings from simultaneous monitoring of rock fracture with AE, GNSS, resistivity, temperature and meteorological measurements? Such comprehensive monitoring is absolutely impossible in another mountains. Short reviews would be useful for the readers working out of Alps.

2. The manuscript length would be shorten. There are some redundant writing in the manuscript, for instance, early parts of '2.1 Research interest at this site', it would be merged with "2. Site description". Please check more for other parts, too.

Best Regards.

---

## Author Comment (AC1) · 29 May 2019

Dear Editor,

We thank for the constructive and positive feedback. The referees state the study as very interesting and further highlight the research as significant. However, they also identified a few minor issues which mainly concern minor language/editorial changes and a comment regarding the inclusion of technology and online data portal specific sections in this data publication. We addressed all the issues raised by the referees and briefly outline here the main revisions we made:

- We corrected all minor editorial/language issues (Reviewer #1).

- We reduced the complexity and improved the readability by shortening and streamlining the technology and data management section (Reviewer #1 and Reviewer #2). The information on the online data portal are now moved to the appendix.

- We reorganized and extended the field site description (Section 2, Reviewer #1).

- We removed duplicates and streamlined the whole text, mainly in sections 1, 2 and 3.

- We added short reviews of the main conclusions found so far based on this data set (Section 6, Reviewer #2).

In the revised manuscript we addressed all the referees' comments and added in the general response one by one explanations (in blue) to the points raised by the referees. If the Editor wishes, we can publish an updated data. However, we would prefer to provide yearly updates at the beginning of a new calendar year.

With kind regards,
Samuel Weber + Jan Beutel
On behalf of all authors

Please also note the supplement to this comment:
https://www.earth-syst-sci-data-discuss.net/essd-2019-14/essd-2019-14-AC1-supplement.pdf

**Supplement:**

**Response to Referee's Comments**
**concerning manuscript *essd-2019-14**

Samuel Weber et al.

May 29, 2019

Dear Editor,

We thank for the constructive and positive feedback. The referees state the study as very interesting and further highlight the research as significant. However, they also identified a few minor issues which mainly concern minor language/editorial changes and a comment regarding the inclusion of technology and online data portal specific sections in this data publication. We addressed all the issues raised by the referees and briefly outline here the main revisions we made:

- We corrected all minor editorial/language issues (Reviewer #1).

- We reduced the complexity and improved the readability by shortening and streamlining the technology and data management section (Reviewer #1 and Reviewer #2). The information on the online data portal are now moved to the appendix.

- We reorganized and extended the field site description (Section 2, Reviewer #1).

- We removed duplicates and streamlined the whole text, mainly in sections 1, 2 and 3.

- We added short reviews of the main conclusions found so far based on this data set (Section 6, Reviewer #2).

In the revised manuscript we addressed all the referees comments and added in the general response one by one explanations (in blue) to the points raised by the referees. If the Editor wishes, we can publish an updated data. However, we would prefer to provide yearly updates at the beginning of a new calendar year.

With kind regards,
Samuel Weber + Jan Beutel
On behalf of all authors

**Reply to comments made by Reviewer Paolo Pogliotti**

We thank Referee Paolo Pogliotti for the review and suggestions for improvement. The response by the authors to the reviewer comments are listed and explained.

**General comment**:  The dataset presented is impressive and of very high quality. I believe the article is excellent for publication in ESSD after a careful reorganization of the contents to make it much leaner and exclusively focused on the Pangaea dataset.

This manuscript presents an impressive dataset recorded over ten years on the north east ridge of Matterhorn, in steep bedrock permafrost conditions.  The dataset includes multi-spot measures of rock temperature and resistivity at different depths, fracture kinematics, displacement and tilting of rock buttresses, time-lapse images and in situ meteorological data.Data collection methods and sensors setup are exhaustively discussed. The dataset, that includes raw data and post-processed products, is openly accessible and available from a long-term data repository (Pangaea).

The dataset is new and extremely relevant for researchers studying the instability processes related to permafrost degradation in high-mountain environment. The variety of measures represents a great added value that allows the use of these data for developing, calibrating or validating process-oriented models as well as designing and testing of remote early-warning systems.  Moreover a number of specific fields of research like e.g. geomatics, hydrology, data-science and more could potentially benefit of this dataset.

To conclude, I believe the article is excellent for publication in ESSD after a careful reorganization of the contents to make it much leaner and exclusively focused on the Pangaea dataset.

**Major comment 1**: The manuscript provides all the informations needed to other researchers for an effective use of the dataset but sometimes these informations are lost in the complexity of the paper. This happens because the article is not exclusively focused on the dataset published in Pangaea but also on the Permasense web portal.  Although the two objects are intimately linked, I believe it is important to keep them separate to promote the use of published data (that can be cited by others) rather than the use of real-time ones (that cannot).  I strongly suggest moving all content related to access and use of the Permasense web portal, included the processing code published on zenodo, to appendix A or publishing it in a separate article.
**Reply**: We thank the referee for this comment.  We agree, the manuscripts contains a lot of information and is very dense.  Nevertheless, these information including complex information as for example the time synchronization are a prerequisite for full reproducibility and understanding of the data and their quality at the raw data level (which we agree is a special use case that not many people will likely be using).  Further, we decided not to focus exclusively on a static data data set (e.g. the current Pangaea version) as the experiment/data acquisition at the Matterhorn Hörnligrat field site are ongoing and all the processing steps and tools are needed for a future update of the data set. Apart from future updates to the dataset by the authors the reader can update the dataset at his discretion using the toolset provided (living data process). We implemented the following changes to improve the readability:  (i) removal of duplicate and confusing parts, (ii) description of the online portal in the appendix only, (iii) a clearer distinction between primary data, secondary data and the processing toolset.

**Major comment 2**:  The Chapter 3  Technology is out of the purpose of this paper because of tailored on the hardware and software architecture of the GSN and the related storage infrastructure.  These aspects are very important and impressive but, in fact, irrelevant for the effective use of the dataset. Also for this, I strongly suggest moving this chapter in appendix A or use it as core for a separate publication in another journal and keep in this publication just the essential (like e.g. figures 2 and 3).

**Reply**: We agree that for the downsampled derived data set in this paper, most of the technology description is not of primary importance. However to be able to understand in full and have transparency and reproducibility as well as in order to be able to leverage the high rate raw primary data set (should that be an objective) it is quite important to know details about the digitizing system and time abstraction used as the instruments used here are not commercial off the shelf (with the then available vendor documentation). It would be possible to move parts of section 3 to the appendix, but at least sections 3.2 and 3.3 on sensor integration, and large parts of section 3.4 data management would be required in the core part of the paper in order to describe the sensor/data specific aspects.

We have therefore opted to move the online access to the appendix but only streamline the sensor technology and data management section.

**Major comment 3**: In the context of environmental science and notably in permafrost studies, the site description (chapter 2) is important and should provide the reader (who does not know the area) with all the basic information, to frame the area in therms of climate, topography, morphology and geology. In my opinion these general informations are totally missing in the manuscript and this gap must be filled. I strongly suggest to provide the following general informations:- climate: temperature, precipitation, seasonal extremes, etc... on average-topography/morphology: mean orientation of the ridge, mean aspect and slopes of the ridge flanks,elevation interval, etc...- geology: main lithologies and rough structural setting (e.g. orientation of the main faults and fractures families...) I guess that in Haslers papers and PhD thesis all these infos are more or less ready to go. See also the technical corrections for further comments.

**Reply**: We fully agree and therefore added the missing information as suggested.

**Major comment 4**: Downloading the dataset from Pangaea repository I noticed that all the .csv files contains a lot of decimal places. In my opinion the dimension of the archives could be significantly reduced by rounding all the double values to the 2nd decimal place. Please consider this.

**Reply**: For some of the derived data this is possible, but for some others, e.g. crackmeters or GNSS strictly rounding to two decimal places would result in reduced fidelity/accuracy of the data. Therefore we suggest to not change this as it would generate inconsistencies in data formatting. E.g.

```
e            n            h          sd_e     sd_n     sd_h
2617909.635 1092124.515 3538.041 0.00011 0.00013 0.00027
```

**Minor comment 1 (P2/R19-21)**: remake this sentence the sense is understandable but not easy readable. Maybe the brackets are not well positioned?
**Reply**: We rephrased this sentence in the revised manuscript

**Minor comment 2 (P2/R24)**: put a comma after setting
**Reply**: Done.

**Minor comment 3 (P2/R25)**: "data: The longest" uppercase after the colon. This is systematic in your paper but Mr. Google told me that is not usual: *'Capitalization: First Word After a Colon. In British English, the first letter after a colon is capitalized only if it's a proper noun or an acronym; in American English, the first word after a colon is sometimes capitalized if it begins a complete sentence'*.
**Reply**: Changed according the suggestion in the revised manuscript.

**Minor comment 4 (P2/R25)**: remove the acronym w.r.t. whole over the paper (lot of occurrences), use the full words instead
**Reply**: Done.

**Minor comment 5 (P2/R31-32)**: this sentence is a kind of repetition, can be removed
**Reply**: Removed.

**Minor comment 6 (P3/R9)**: what do you mean with sensor (type) extension?
**Reply**: This section as been removed and consolidated as outlined for major comment 1 and 2.

**Minor comment 7 (P3/R21-32:)**: move & merge this block of text in the research interest-paragraph 2.1.See also major comments n.3
**Reply**: Done. Section 2 has been consolidated.

**Minor comment 8 (P4/R4)**: please provide a most statistically significant MAAT, at least over the period 2008-2018. If possible provide also the mean annual precipitation (mm), seasonal extremes and other useful climatic informations. See also major comments n.3
**Reply**: Done. Since the weather station has experienced serious data gaps and was only installed mid through the whole monitoring period it is not so straightforward to calculate a correct MAAT from our data. Therefore we have given one derived from literature and an example value for a period where full data from our instruments exist.

**Minor comment 9 (P5/R22-25)**: 'All the ... context', already said at the end of the Chapter 1-Introduction... remove or merge there.
**Reply**: This has been removed here, the end of the intro was reformulated.

**Minor comment 10 (P6-P11)**: See major comments n.1&2
**Reply**: This sentence has been shortened to only contain the sensors/features that are part of this dataset/paper.

**Minor comment 11 (P12/R16-18)**: a description of section 5 is not pertinent here. Move it to the end of the chapter 1 or delete.
**Reply**: In our opinion, this statement is important to remind the reader that the derived data product is described in the next section. We moved the details to the end of the introduction as suggested.

**Minor comment 12 (P12/R19)**: It would be very useful to have the figure 16 in this chapter, between table 2 and figure 6. This is important especially for the comprehension of caption in table 2 and remarks in rows 19-26.
**Reply**: Done.

**Minor comment 13 (P12/R20)**: 'reliability in place of 'availability'
**Reply**: Done.

**Minor comment 14 (P12/R23-24)**: 'Therefore... paper' is not necessary. This remark already rise up from previous sentences.
**Reply**: Removed.

**Minor comment 15 (P12/R28-32)**: Remove the first sentence that is well-known and move this block at the end of paragraph. That is, start the paragraph with 'Since 2010...'
**Reply**: This paragraph was moved to a separate subsection in section 4.8 further data and related work.

**Minor comment 16 (P13/R1)**: in Table 1 WXT520/536 is cited, while in text just WXT520. In fact WXT536 appears just in tab.1. Please fix this inconsistency or explain it.
**Reply**: The WXT536 is an updated variant of the WXT520 product. It delivers the same data although some technical (interface) details differ. We removed the reference to WXT536.

**Minor comment 17 (P15/R7)**: define near surface in therms of depth... 10cm?
**Reply**: We defined *near-surface temperature* as measurement at $3 - 8\,\mathrm{cm}$ depth.

**Minor comment 18 (P15/R8)**: explain the acronym NTC
**Reply**: Done.

**Minor comment 19 (P15/R12)**: please, add a reference to Tab.1 after 'used'
**Reply**: Done.

**Minor comment 20 (P15/R12)**: remove the last sentence, it is a repetition of row P15/R8-9
**Reply**: Done.

**Minor comment 21 (P16/Fig.7)**: this figure is not cited in text (notably paragraph 4.1). Anyway I guess it is not very relevant, I let you decide whether to keep it or take it off
**Reply**: We changed the manuscript accordingly and now refer to the figure in the revised manuscript.

**Minor comment 22 (P16/R4)**: explain acronym ADC. Please add the drift values also for this sensor (if possible)
**Reply**: ADC is corrected. The manufacturer does not supply drift information.

**Minor comment 23 (P16/R10)**: Table 2 shows the depths of temperature measures in rock face and fractures but the indication of which of the 4 thermistor systems is used is missing.To address this task you can label (as e.g. A,B,C,D) the ground temperature in Tab.1 (first column) and text in this paragraph (P16/R1-9), then call the same labels in the caption of Tab.2 like e.g.: [a]Intervention: Change of thermistor system from A to D. [b]Intervention: Change of thermistor system from B to C. etc... Tab.2 might become little more complex but more exhaustive and meaningful.
**Reply**: Done.

**Minor comment 24 (P17/Fig.8 and 9)**: due to the length of the time series, evaluate of enlarging the plots exploiting the entire width of the page while keeping fix the present height
**Reply**: We addressed this comment in the revised manuscript and changed Figure 7 to Figure 10 accordingly.

**Minor comment 25 (P19/Fig.10)**: same as above
**Reply**: Done. See reply to minor comment 24.

**Minor comment 26 (P19/R2)**: bis → is. This sentence could be replicated in the tables caption.
**Reply**: Done.

**Minor comment 27 (P19/R4)**: remove 'here'
**Reply**: Done.

**Minor comment 28 (P19/R6)**: add a comma after 'rockfall'
**Reply**: Done.

**Minor comment 29 (P19/R13)**: remove the last 'remotely'
**Reply**: Done.

**Minor comment 30 (P19/R14)**: comma after 'active'
**Reply**: Done.

**Minor comment 31 (P20/R2–4)**: please, indicate clearly in the first sentence that what you call the primary antenna, works as a master station with respect to the others and that it is

located at point MH42/HOGR. At first glance, it was not easy to be sure of that. I know this will be discussed further in section 5.2, but it is better to have it here too.
**Reply**: Done.

**Minor comment 32 (P22/R17-18)**: check the English, the sentence sounds strange
**Reply**: This section has been reworded in accordance with major comments 1 and 2.

**Minor comment 33 (P22/R22-23)**: This sentence can be omitted
**Reply**: Minor rewording done for clarity.

**Minor comment 34 (P30/R24)**: use acoustic emission / microseismic instead of AE/MS
**Reply**: Done.

**Minor comment 35 (P31/R1)**: See major comments n. 1 & 2
**Reply**: Done.

**Minor comment 36 (P32/Table A1)**: The reference to this table is missing in the text.
**Reply**: Done.

**Reply to comments made by Anonymous Reviewer #2**

We thank Anonymous Referee #2 for the review and suggestions for improvement. The response by the authors to the reviewer comments are listed and explained.

**General comment**: This is very excellent manuscript since it showcases comprehensive and state-of-the-arts data sets recording thermal, hydraulic and kinematic dynamics of permafrost underlying alpine bedrock cliffs where few observations have been conducted in the world due to its harsh environments. Manuscript is also written very well with perfect information of technological and geographical settings, and data quality, accordingly I strongly recommend it to be published ESSD after minor revisions.

There are two points that I would like to comment for further revisions.

**Major comment 1**: '6. Example of data use': In this section, the authors only listed some publications (phD thesis,etc., ) based on this data sets. I recommend to describe here more scientifically, e.g.,what was new scientific findings from simultaneous monitoring of rock fracture with AE, GNSS, resistivity, temperature and meteorological measurements? Such comprehensive monitoring is absolutely impossible in another mountains. Short reviews would be useful for the readers working out of Alps.
**Reply**: Done. We included more context here.

**Major comment 2**: The manuscript length would be shorten. There are some redundant writing in the manuscript, for instance, early parts of 2.1 Research interest at this site, it would be merged with "2. Site description". Please check more for other parts, too.
**Reply**: We have substantially streamlined sections 1, 2 and 3 and removed duplicate discussions in various other part. See also the answers to the major comments of reviewer one.

---

## Author Response (AR2)

**Response to Referee's Comments**
**concerning manuscript *essd-2019-14**

Samuel Weber et al.

July 4, 2019

Dear Editor,

We thank for the positive feedback. We addressed the suggested minimal technical corrections of Paolos in the revised manuscript.

With kind regards,
Samuel Weber + Jan Beutel
On behalf of all authors

[revised manuscript text omitted]